# The ESCRT protein CHMP5 restricts bone formation by controlling endolysosome-mitochondrion-mediated cell senescence

Fan Zhang[1,2,3†], Yuan Wang[1,2†], Luyang Zhang[1,2], Chunjie Wang[1,2], Deping Chen[4], Haibo Liu[5], Ren Xu[6], Cole M Haynes[5], Jae-Hyuck Shim[7*], Xianpeng Ge[1,2*]

[1]Xuanwu Hospital Capital Medical University, Beijing, China; [2]National Clinical Research Center for Geriatric Diseases, Beijing, China; [3]Joint Therapeutics Co. Ltd, Beijing, China; [4]Beijing Citident Hospital of Stomatology, Beijing, China; [5]Department of Molecular, Cell and Cancer Biology, University of Massachusetts Chan Medical School, Worcester, United States; [6]The First Affiliated Hospital of Xiamen University-ICMRS Collaborating Center for Skeletal Stem Cells, State Key Laboratory of Cellular Stress Biology, Faculty of Medicine and Life Sciences, School of Medicine, Xiamen University, Xiamen, China; [7]Department of Medicine/Division of Rheumatology, University of Massachusetts Chan Medical School, Worcester, United States

*For correspondence:
jaehyuck.shim@umassmed.edu (J-HS);
xianpeng.ge@xwhosp.org (XG)

†These authors contributed equally to this work

## eLife Assessment

This **important** work advances our understanding of CHMP5's role in regulating osteogenesis through its impact on cellular senescence. The evidence supporting the conclusion is **convincing** and the revised manuscript is largely improved. This paper holds potential interest for skeletal biologists who study the pathogenesis of age-associated skeletal disorders.

**Abstract** The dysfunction of the cellular endolysosomal pathway, such as in lysosomal storage diseases, can cause severe musculoskeletal disorders. However, how endolysosomal dysfunction causes musculoskeletal abnormalities remains poorly understood, limiting therapeutic options. Here, we report that CHMP5, a member of the endosomal sorting complex required for transport (ESCRT)-III protein family, is essential to maintain the endolysosomal pathway and regulate bone formation in osteogenic lineage cells. Genetic ablation of *Chmp5* in mouse osteogenic cells increases bone formation in vivo and in vitro. Mechanistically, *Chmp5* deletion causes endolysosomal dysfunction by decreasing the VPS4A protein, and CHMP5 overexpression is sufficient to increase the VPS4A protein. Subsequently, endolysosomal dysfunction disturbs mitochondrial functions and increases mitochondrial ROS, ultimately resulting in skeletal cell senescence. Senescent skeletal cells cause abnormal bone formation by combining cell-autonomous and paracrine actions. Importantly, the elimination of senescent cells using senolytic drugs can alleviate musculoskeletal abnormalities in *Chmp5* conditional knockout mice. Therefore, our results show that cell senescence represents an underpinning mechanism and a therapeutic target for musculoskeletal disorders caused by the aberrant endolysosomal pathway, such as in lysosomal storage diseases. These results also uncover the function and mechanism of CHMP5 in the regulation of cell senescence by affecting the endolysosomal-mitochondrial pathway.

## Introduction

The endosomal sorting complex required for transport (ESCRT) machinery is an evolutionarily conserved molecular mechanism essential for diverse cell biological processes. Since it was initially discovered in the early 21st century in the budding yeast *Saccharomyces cerevisiae*, the molecular gear has been extensively dissected in the endocytic pathway, which sorts ubiquitinated membrane proteins and bound cargoes that commit to lysosomal degradation (*Kaksonen and Roux, 2018*; *Christ et al., 2017*; *Luzio et al., 2007*; *Hurley, 2015*). In addition to the endocytic process, ESCRT proteins also play critical roles in other cell activities, such as exocytosis, viral budding, cytokinesis, pruning of neurons, repair of the plasma membrane, and assembly of the nuclear envelope (*Christ et al., 2017*; *Hurley, 2015*). In the endocytic pathway, the ESCRT machinery consists of a series of functionally overlapping protein complexes, including ESCRT-0, -I, -II, and -III. While the ESCRT-0, -I, and -II complexes are mainly responsible for protein sorting and recruiting ESCRT-III components, the ESCRT-III complex, cooperating with the ATPase VPS4, is the primary executor of membrane severing during the formation of multivesicular bodies (MVB) (*Luzio et al., 2007*; *Schmidt and Teis, 2012*), which ultimately fuse with the lysosomes for cargo degradation or with the plasma membrane to release cargos into the extracellular space.

Charged multivesicular body protein 5 (CHMP5), the mammalian ortholog of yeast VPS60/MOS10, is a component of the ESCRT-III protein complex and plays an essential role in the late stage of MVB formation (*Shim et al., 2006*). As the yeast *Vps60/Mos10*-null mutation disturbs the endosome-to-vacuole (the lysosome-like structure in yeast) trafficking and affects the sorting of internalized endocytic markers (*Kranz et al., 2001*), ablation of *Chmp5* in mouse embryonic fibroblasts (MEFs) also affects the sorting from late endosomes/MVBs to lysosomes and reduces lysosomal degradation of multiple activated receptors (*Shim et al., 2006*). Notably, in mouse osteoclasts, CHMP5 suppresses ubiquitylation of the IκBα protein and restricts the activity of the NFκB pathway, without any effect on endolysosomal functions (*Greenblatt et al., 2015*). During αβ T lymphocyte development, CHMP5 promotes thymocyte survival by stabilizing the pro-survival protein of BCL2 and does not have an influence on the endolysosomal system (*Adoro et al., 2017*). These studies suggest that the regulation of the endolysosomal pathway by CHMP5 depends on the cell and tissue context.

The dysfunction of the endolysosomal pathway can cause severe musculoskeletal disorders. A such example is lysosomal storage diseases, which include more than 50 different subgroups of diseases due to mutations in genes that encode lysosomal hydrolases or genes that are responsible for the transport, maturation, and functions of lysosomal hydrolases (*Parenti et al., 2021*). Musculoskeletal pathologies, such as joint stiffness/contracture, bone deformation, muscular atrophy, short stature, and decreased mobility, are among the most common, in some cases the earliest, clinical presentations of lysosomal storage diseases (*Morishita and Petty, 2011*; *Clarke and Hollak, 2015*). These musculoskeletal lesions are particularly refractory to current treatments for lysosomal storage diseases, including enzyme replacement treatment and hematopoietic stem cell transplantation (*Miller et al., 2023*). Therefore, studies on the mechanisms of musculoskeletal pathologies due to endolysosomal dysfunction are paramount to design and develop novel treatments for these disorders.

In this study, our results uncover the function and mechanism of CHMP5 in the regulation of cell senescence and bone formation in osteogenic cells. Deletion of *Chmp5* causes endolysosomal dysfunction involving decreased VPS4A protein and subsequently activates cell senescence by increasing intracellular mitochondrial ROS. Senescent cells induce bone formation in both autonomous and paracrine ways. Importantly, senolytic treatment is effective in mitigating musculoskeletal pathologies caused by *Chmp5* deletion.

## Results

### Ablation of *Chmp5* in mouse osteogenic lineage cells causes aberrant bone formation

In addition to the well-characterized role of the CTSK gene in osteoclasts, this gene has recently been reported to identify periskeletal progenitors (*Yang et al., 2013*; *Debnath et al., 2018*; *Shi et al., 2018*). In particular, the ablation of *Chmp5* in mouse *Ctsk*-expressing cells (hereafter referred to as *Ctsk*<sup>Cre</sup>;*Chmp5*<sup>fl/fl</sup> mice, homozygous) caused dramatic periskeletal bone overgrowth near the joint compared to *Ctsk*<sup>Cre</sup>;*Chmp5*<sup>fl/+</sup> (heterozygous) and *Chmp5*<sup>fl/fl</sup> littermate controls (*Figure 1A and B*).

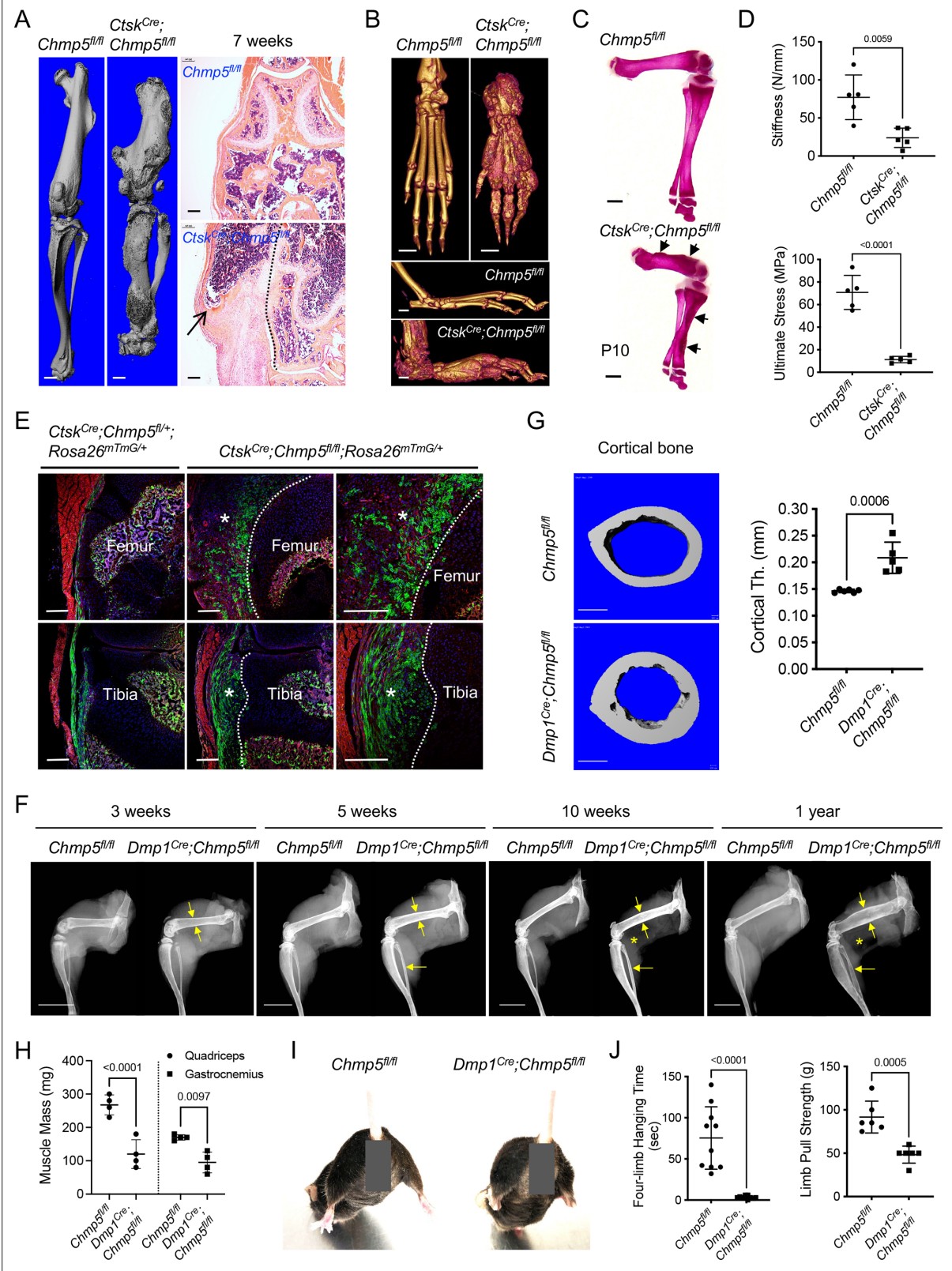

**Figure 1.** Ablation of charged multivesicular body protein 5 (*Chmp5*) in mouse osteogenic lineage cells causes aberrant bone formation. (**A**) Micro-CT and H&E staining showing periskeletal overgrowth in *Ctsk^Cre^;Chmp5^fl/fl^* mice in comparison with *Chmp5^fl/fl^* mice at 7 wk of age. Dot line representing the approximate bone border. n=10 animals per group. (**B**) Micro-CT images displaying periskeletal overgrowth in the ankle and foot in *Ctsk^Cre^;Chmp5^fl/ fl^* versus *Chmp5^fl/fl^* mice at 1 y of age. n=4 animals per group. (**C**) Alizarin red staining of skeletal preparations on postnatal day 10. n=4 mice per group,

*Figure 1 continued on next page*

*Figure 1 continued*

arrows indicating periskeletal bone overgrowth. (**D**) Three-point test showing femur bone stiffness and ultimate stress (fracture stress) in *Ctsk^Cre^;Chmp5^fl/fl^* and *Chmp5^fl/fl^* mice. n=5 male animals per group. Similar changes were observed in both sexes. (**E**) Confocal images mapping *Ctsk^+^* (GFP^+^) progenitors in periskeletal tissues in *Ctsk^Cre^;Chmp5^fl/+^;Rosa26^mTmG/+^* and *Ctsk^Cre^; Chmp5^fl/fl^;Rosa26^mTmG/+^* mice at 2 wk of age. Asterisks indicate periskeletal overgrowths; dot line represents the approximate bone border. n=4 animals each group. (**F**) X-ray images demonstrating progressive cortical bone expansion (arrows) and decreased skeletal muscle mass (asterisks) in *Dmp1^Cre^;Chmp5^fl/fl^* in comparison with *Chmp5^fl/fl^* mice. n=3, 3, 10, 8 animals per group for 3 wk, 5 wk, 10 wk, and 1 y of age, respectively. (**G**) Micro-CT analyses displaying cortical bone expansion in *Dmp1^Cre^;Chmp5^fl/fl^* relative to *Chmp5^fl/fl^* mice at 10 wk of age. n=6 *Chmp5^fl/fl^* and 5 *Dmp1^Cre^;Chmp5^fl/fl^* male mice, similar changes found in both genders. (**H**) Skeletal muscle mass in *Dmp1^Cre^;Chmp5^fl/fl^* compared to *Chmp5^fl/fl^* mice at 10–12 wk of age. n=4 animals (male) per group, similar changes found in both genders. (**I**) Hindlimb abduction of *Dmp1^Cre^;Chmp5^fl/fl^* mice in comparison with *Chmp5^fl/fl^* littermate controls at 10 wk of age. More than 20 animals per group were observed. (**J**) Four-limb handing time (n=10 per group, pooled data from both sexes), and forelimb pull strength (n=6 per group, male mice) in *Dmp1^Cre^;Chmp5^fl/fl^* compared to *Chmp5^fl/fl^* mice at 10–12 wk of age. Similar changes found in both genders. All data are mean ± s.d.; two-tailed unpaired Student's *t*-test. Scale bars, 1 mm for micro-CT and skeletal preparation images, 200 μm for histological images, and 10 mm in panel (**F**).

The online version of this article includes the following source data and figure supplement(s) for figure 1:

**Source data 1.** Data of three-point test, cortical bone thickness, and quantification of skeletal muscle mass and functions.

**Figure supplement 1.** Charged multivesicular body protein 5 (CHMP5) restricts bone formation in osteogenic lineage cells.

**Figure supplement 1—source data 1.** For *Figure 1—figure supplement 1D*, Data of whole femur thickness (mm) in *Dmp1^Cre^;Chmp5^fl/fl^* versus *Chmp5^fl/fl^* mice at 10 wk of age.

Aberrant periskeletal bone growth in *Ctsk^Cre^;Chmp5^fl/fl^* mice became distinct around postnatal day 10 (d10), progressed with age, and involved multiple bones and joints throughout the body, including knee, ankle, and foot joints (*Figure 1A–C* and *Figure 1—figure supplement 1A*). With age, these animals developed severe skeletal deformities, progressive joint stiffness, reduced motility, and short stature. Despite bone overgrowth in *Ctsk^Cre^;Chmp5^fl/fl^* animals, mechanical tests showed that bone stiffness and fracture stress decreased significantly (*Figure 1D*), suggesting that bone mechanical properties were impaired in *Ctsk^Cre^;Chmp5^fl/fl^* mice.

To track CTSK^+^ periskeletal progenitors during the bone overgrowth in *Ctsk^Cre^;Chmp5^fl/fl^* mice, we crossed *Ctsk-Cre*, *Chmp5^fl/fl^*, and *Rosa26^mTmG^* mice to generate *Ctsk^Cre^;Chmp5^fl/+^;Rosa26^mTmG/+^* and *Ctsk^Cre^;Chmp5^fl/fl^;Rosa26^mTmG/+^* reporter animals, in which CTSK^+^ periskeletal progenitors and their descendants express membrane-localized green fluorescence protein (GFP). In *Ctsk^Cre^;Chmp5^fl/+^; Rosa26^mTmG/+^* control mice, GFP^+^ cells could be detected in the perichondrium, groove of Ranvier, periosteum, ligament, and enthesis in addition to osteoclasts in the bone marrow, but not in growth plate chondrocytes (*Figure 1E*). With *Chmp5* deletion, GFP^+^ cells consisted of a major part of the cells in periskeletal overgrowth in *Ctsk^Cre^;Chmp5^fl/fl^; Rosa26^mTmG/+^* mice (*Figure 1E*, *Figure 1—figure supplement 1B*). Tartarate-resistant acid phosphatase (TRAP) staining did not show TRAP^+^ osteoclasts in the periskeletal lesion, while obvious TRAP^+^ osteoclasts could be detected in the bone marrow (*Figure 1—figure supplement 1C*). These results indicate that *Chmp5*-deficient periskeletal progenitors directly contribute to the bone overgrowth in *Ctsk^Cre^;Chmp5^fl/fl^* mice.

Next, we sought to confirm the role of CHMP5 in the regulation of bone formation by deleting this gene using another mouse line of osteogenic lineage Cre. We crossed *Chmp5^fl/fl^* mice with the line of *Prrx1-Cre* (*Logan et al., 2002*), *Col2a1-Cre* (*Ono et al., 2014*) or *Osx-Cre* (*Maes et al., 2010*) mice to delete the *Chmp5* gene in *Prrx1*-expressing limb mesenchymal progenitors (*Prrx1^Cre^;Chmp5^fl/fl^* mice), *Col2a1*-expressing perichondral progenitors (*Col2a1^Cre^; Chmp5^fl/fl^* mice) or *Osx*-expressing osteoprogenitors (*Osx^Cre^;Chmp5^fl/fl^* mice) and were unable to obtain postnatal homozygous gene knockout mice, probably due to embryonic lethality after ablating C*hmp5* expression in these osteogenic lineages. Subsequently, we used the *Dmp1-Cre* mouse line that could target postnatal endosteal osteoprogenitors, preosteoblasts, mature osteoblasts, and osteocytes (*Lim et al., 2016*; *Lim et al., 2017*; *Matic et al., 2016*). *Dmp1^Cre^;Chmp5^fl/fl^* mice were born in normal Mendelian ratios and were viable. Within 3 wk of age, the skeletons of *Dmp1^Cre^;Chmp5^fl/fl^* mice began to show a distinct expansion in comparison with *Dmp1^Cre^;Chmp5^fl/+^* and *Chmp5^fl/fl^* littermate controls, which progressed with age and became much more severe at 1 y of age (*Figure 1F*). Bone expansion in *Dmp1^Cre^;Chmp5^fl/fl^* mice that occurred mainly in the endosteum was further shown in detail around the age of 10 wk by micro-CT and histological analyses (*Figure 1G* and *Figure 1—figure supplement 1D, E*).

Meanwhile, since *Dmp1*-expressing lineage cells also generate skeletal muscle cells (*Lim et al., 2017*), *Dmp1^Cre^;Chmp5^fl/fl^* mice showed a profound decrease in skeletal muscle mass with age

(*Figure 1F and H*). Hindlimb abduction, full-limb hanging time, and front-limb pull strength tests showed that muscular functions also decreased profoundly in *Dmp1^Cre^;Chmp5^fl/fl^* compared to *Chmp5^fl/fl^* mice (*Figure 1I and J*).

## CHMP5 restricts osteogenesis in skeletal progenitor cells

To further characterize the function of CHMP5 in the regulation of bone formation, we cultured and sorted *Ctsk*-expressing periskeletal progenitor cells (CD45⁻;CD31⁻;GFP⁺) from periskeletal tissues of *Ctsk^Cre^;Chmp5^fl/+^;Rosa26^mTmG/+^* and *Ctsk^Cre^;Chmp5^fl/fl^; Rosa26^mTmG/+^* mice in the early stage of skeletal lesions (P10-P14) (*Figure 2A* and *Figure 2—figure supplement 1A, B*, hereafter referred to as *Ctsk^Cre^;Chmp5^fl/+^* and *Ctsk^Cre^;Chmp5^fl/fl^* periskeletal progenitors, respectively). *Chmp5* deletion in *Ctsk^Cre^;Chmp5^fl/fl^* periskeletal progenitors was confirmed by gene expression analysis (*Figure 2B*). Strikingly, when cultured in the osteogenic differentiation medium, *Ctsk^Cre^;Chmp5^fl/fl^* periskeletal progenitors showed considerably enhanced osteogenesis in comparison with *Ctsk^Cre^;Chmp5^fl/+^* control cells, as shown by alizarin red staining, von Kossa staining, and alkaline phosphatase activity assay (*Figure 2C*).

The regulation of osteogenesis by CHMP5 in skeletal progenitors was further determined in the osteogenic cell line MC3T3-E1 after removing *Chmp5* using CRISPR/CAS9 technology (*Figure 2D*). The deletion of *Chmp5* in MC3T3-E1 cells also markedly increased osteogenesis, as shown by alizarin red staining at 2 wk and 3 wk of osteogenic induction (*Figure 2E*). Consistently, the ingenuity pathway analysis of RNA-seq data of *Ctsk^Cre^;Chmp5^fl/fl^* vs. *Ctsk^Cre^;Chmp5^fl/+^* periskeletal progenitors showed that multiple molecular pathways related to osteoblast differentiation were activated with the *Chmp5* deletion (*Figure 2F*). Taken together, these results reveal a function of CHMP5 in regulating osteogenesis in skeletal progenitors.

## CHMP5 controls skeletal progenitor cell senescence

To gain more insight into the mechanism of regulation of bone formation by CHMP5, we assumed that depletion of *Chmp5* in osteogenic lineage cells also increases cell proliferation, which contributes to the bone overgrowth in *Chmp5* conditional knockout mice. Unexpectedly, *Ctsk^Cre^;Chmp5^fl/fl^* periskeletal progenitors showed a significant decrease in the cell proliferation rate compared to *Ctsk^Cre^;Chmp5^fl/+^* controls (approximately 58% decrease in the cell number on day 6 of P2 culture, *Figure 3A*). Because CHMP5 has been reported to regulate thymocyte apoptosis (*Adoro et al., 2017*), we performed cell apoptosis analyses for *Ctsk^Cre^;Chmp5^fl/fl^* and *Ctsk^Cre^;Chmp5^fl/+^* cells. Annexin V staining showed that approximately $4.96 \pm 0.81\%$ of *Ctsk^Cre^;Chmp5^fl/fl^* periskeletal progenitors vs. $1.69 \pm 0.68\%$ of *Ctsk^Cre^;Chmp5^fl/+^* control cells were positively stained (*Figure 3—figure supplement 1A*). Similarly, the in-situ terminal deoxynucleotidyl transferase dUTP nick end labelling assay (TUNEL) demonstrated that around $3.70 \pm 0.72\%$ cells in the periskeletal lesion of *Ctsk^Cre^;Chmp5^fl/fl^* mice were positively labelled (*Figure 3—figure supplement 1B*). These results show that cell apoptosis is activated in *Ctsk^Cre^;Chmp5^fl/fl^* periskeletal progenitors in a rather low ratio.

Notably, the low apoptotic cell ratio (approximately 5%) could not well match the decrease in cell proliferation rate (approximately 58%) in *Ctsk^Cre^;Chmp5^fl/fl^* periskeletal progenitors. We further analyzed the RNA-seq data of *Ctsk^Cre^;Chmp5^fl/fl^* vs. *Ctsk^Cre^;Chmp5^fl/+^* cells to dig out the mechanism of CHMP5 deficiency on the fate of skeletal progenitors. The gene set enrichment analysis (GSEA) showed significant enrichment of genes in multiple Reactome molecular pathways associated with cell senescence in *Ctsk^Cre^;Chmp5^fl/fl^* relative to *Ctsk^Cre^;Chmp5^fl/+^* periskeletal progenitors, including *Hmga1*, *Hmga2*, *Trp53*, *Ets1*, and *Txn1* (*Figure 3B and C*). Meanwhile, GSEA of RNA-seq data also showed significant enrichment of the SAUL_SEN_MAYO geneset (positively correlated with cell senescence) and the KAMMINGA_SENESCENCE geneset (negatively correlated with cellular senescence) in *Ctsk^Cre^;Chmp5^fl/fl^* vs. *Ctsk^Cre^;Chmp5^fl/+^* periskeletal progenitors (*Figure 3—figure supplement 1C*).

Furthermore, Western blot results showed that both the p16 and p21 proteins, two canonical markers of cellular senescence, were upregulated in *Ctsk^Cre^;Chmp5^fl/fl^* compared to control cells, with a higher degree of upregulation in the p16 protein (*Figure 3D*). However, the mRNA levels of *Cdkn2a* (p16) and *Cdkn1a* (p21) did not show significant changes according to the RNA-seq analysis (*Figure 3—figure supplement 1D*). Furthermore, immunostaining for another cell senescence marker γH2Ax demonstrated that there were significantly more γH2Ax⁺;GFP⁺ cells in periskeletal overgrowth in *Ctsk^Cre^;Chmp5^fl/fl^;Rosa26^mTmG/+^* mice relative to the periosteum of *Ctsk^Cre^;Chmp5^fl/+^; Rosa26^mTmG/+^*

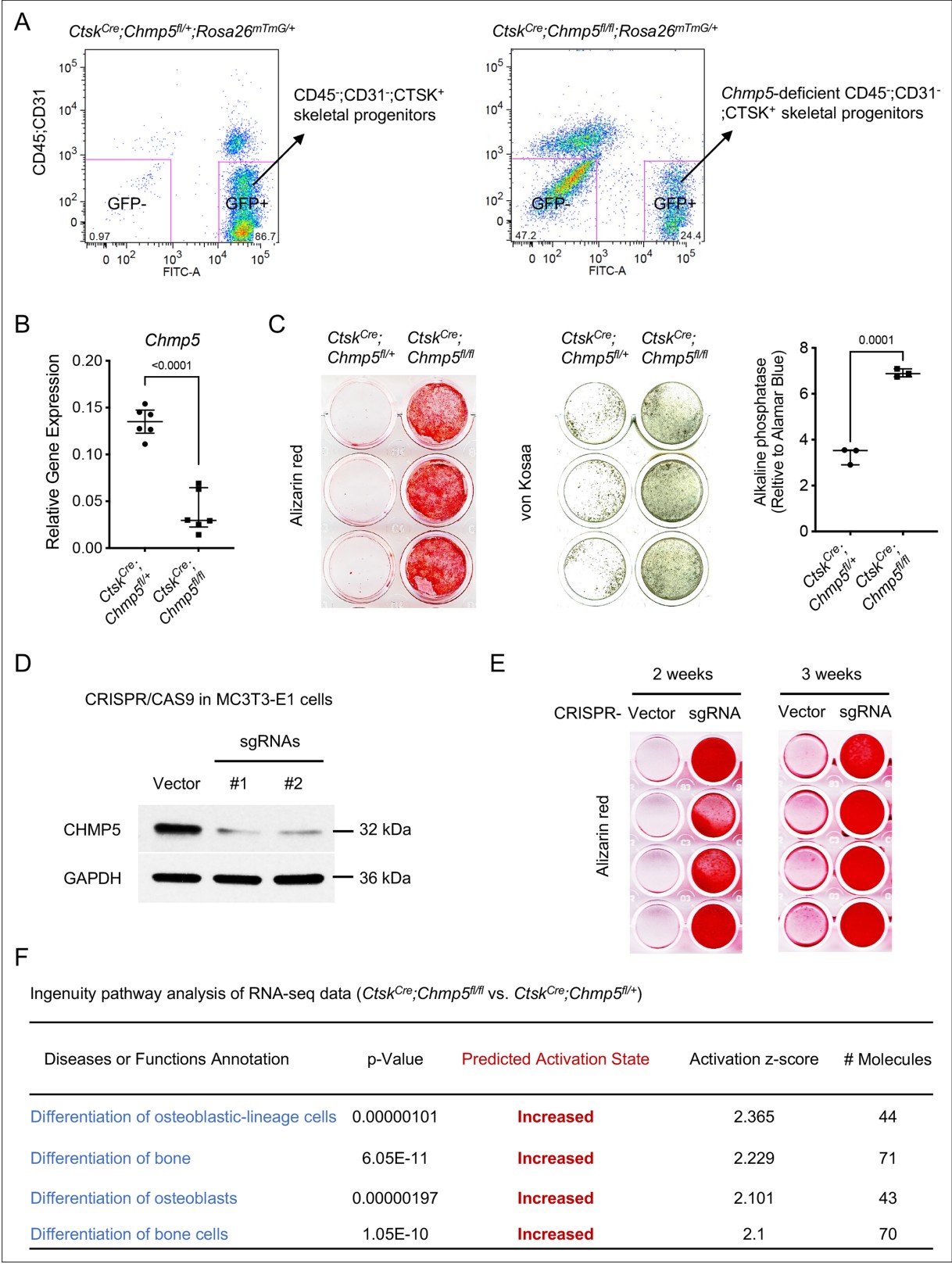

**Figure 2.** Charged multivesicular body protein 5 (CHMP5) restricts osteogenesis in skeletal progenitor cells. (**A**) Flow cytometry showing the strategy for sorting CD45⁻;CD31⁻;GFP⁺ and CD45⁻;CD31⁻;GFP⁻ periskeletal progenitors from periskeletal tissues of *Ctsk^Cre^;Chmp5^fl/+^;Rosa26^mTmG/+^* and *Ctsk^Cre^;Chmp5^fl/fl^;Rosa26^mTmG/+^* mice at 2 wk of age. n=10 mice per group. (**B**) Quantitative PCR determining the expression of *Chmp5* in *Ctsk^Cre^;Chmp5^fl/fl^* relative to *Ctsk^Cre^;Chmp5^fl/+^* periskeletal progenitors. n=6 mice per group. (**C**) Alizarin red staining, von Kossa staining, and alkaline phosphatase activity assay

*Figure 2 continued on next page*

*Figure 2 continued*

determining osteogenesis in *Ctsk^Cre;Chmp5^fl/fl* compared to *Ctsk^Cre;Chmp5^fl/+* periskeletal progenitors. n=3 replicates each group, representative results of cells from three different mice per group. (**D**) Western blot confirming *Chmp5* deletion in mouse MC3T3-E1 cells by lentiviral CRISPR/CAS9. (**E**) Alizarin red staining demonstrating osteogenesis in MC3T3-E1 cells with or without *Chmp5* deletion by lentiviral CRISPR/CAS9. n=4 replicates per group, experiments repeated twice for each time point. (**F**) Ingenuity pathway analysis of RNA-seq data showing increased activity of osteoblast differentiation in *Ctsk^Cre;Chmp5^fl/fl* vs. *Ctsk^Cre;Chmp5^fl/+* periskeletal progenitors. All data are mean ± s.d.; two-tailed unpaired Student's *t*-test.

The online version of this article includes the following source data and figure supplement(s) for figure 2:

**Source data 1.** Data of qPCR and alkaline phosphatase activity assay.

**Source data 2.** Original files for western blot analysis displayed in *Figure 2D*.

**Source data 3.** PDF file containing original western blots for *Figure 2D*, indicating the relevant bands.

**Figure supplement 1.** Sorting of CD45⁻;CD31⁻;CTSK⁺ periskeletal progenitors.

**Figure supplement 1—source data 1.** For *Figure 2—figure supplement 1B*, Data of quantification of CD45⁻;CD31⁻;GFP⁺ and CD45⁻;CD31⁻;GFP⁻ cell population.

control mice (*Figure 3E*). Accordingly, *Ctsk^Cre;Chmp5^fl/fl* mice showed gross accelerated aging-related phenotypes, including hair loss, joint stiffness/contracture, and decreased motility (*Figure 3F*).

To further characterize the effect of CHMP5 on skeletal progenitor cell fate, we deleted *Chmp5* in ATDC5 cells, which were shown to be an appropriate model of periskeletal progenitors in a previous study (*Ge et al., 2016*), using CRISPR/CAS9 technology (*Figure 3—figure supplement 1E*). Similarly, *Chmp5* deletion also significantly suppressed the cell proliferation rate of ATDC5 cells and mildly increased cell apoptosis (*Figure 3—figure supplement 1F, G*). However, inhibition of cell apoptosis using the pancaspase inhibitor Q-VD-Oph could not reverse cell number reduction (*Figure 3—figure supplement 1H*), indicating that cell apoptosis was not the main cause of the decrease in the proliferation rate of *Chmp5*-deficient skeletal progenitors. Instead, cell cycle analysis showed a significant decrease of *Chmp5*-deficient ATDC5 cells in phase S, and a significant proportion of these cells were arrested in G0/G1 or G2/M phases (*Figure 3G*), which is a typical characteristic of cell cycle arrest in senescent cells. Taken together, these results at the molecular, cellular, and gross levels demonstrate that *Chmp5* deficiency induces skeletal progenitor cell senescence.

## Secretory phenotype of *Chmp5*-deficient skeletal progenitors

A remarkable feature of senescent cells is the senescence-associated secretory phenotype (SASP), which plays a critical role in mediating the pathophysiological processes of cell senescence (*Birch and Gil, 2020*). In particular, periskeletal overgrowths in *Ctsk^Cre;Chmp5^fl/fl;Rosa26^mTmG/+* mice showed evident integration of many cells lacking evidence of Cre-mediated recombination (GFP⁻) (*Figure 1E*), suggesting that *Chmp5*-deleted periskeletal progenitors might recruit neighboring wild-type cells to facilitate the bone overgrowth. Additionally, the GSEA of RNA-seq data showed a significant enrichment of genes associated with the SASP molecular pathway in *Chmp5*-deleted periskeletal progenitors (*Figure 4A*). Meanwhile, confocal fluorescence microscopy captured many extracellular vesicles shed from the plasma membrane of *Ctsk^Cre;Chmp5^fl/fl* periskeletal progenitors, which were rarely detected around wild-type control cells (*Figure 4B*). Furthermore, nanoparticle tracking analysis demonstrated higher concentrations of extracellular vesicles in the culture medium of *Ctsk^Cre;Chmp5^fl/fl* periskeletal progenitors compared to wild-type control cells (*Figure 4C, D*, *Figure 4—figure supplement 1A*). These results show that *Chmp5* deficiency activates the SASP molecular pathway and increases the secretion of skeletal progenitors.

Functionally, conditioned medium collected from *Chmp5*-deficient skeletal progenitors caused a higher proliferation rate of ATDC5 cells than medium from control cells (*Figure 4E*). Meanwhile, coculture of wild-type with *Ctsk^Cre;Chmp5^fl/fl* periskeletal progenitors promoted osteogenesis of wild-type cells (*Figure 4—figure supplement 1B*). Simultaneously, CD45⁻;CD31⁻;GFP⁻ skeletal progenitors from periskeletal tissues of *Ctsk^Cre;Chmp5^fl/fl;Rosa26^mTmG/+* mice showed increased proliferation along with enhanced osteogenic differentiation compared to the corresponding cells from *Ctsk^Cre;Rosa26^mTmG/+* control mice (*Figure 4F, G* and *Figure 4—figure supplement 1C, D*). Immunostaining for the cell proliferation marker Ki-67 demonstrated an increase in the number of stained cells in the periskeletal lesion of *Ctsk^Cre;Chmp5^fl/fl* mice (*Figure 4—figure supplement 1E*). Since GFP⁺ *Ctsk^Cre;Chmp5^fl/fl* periskeletal progenitors showed a decreased cell proliferation rate and increased cell senescence

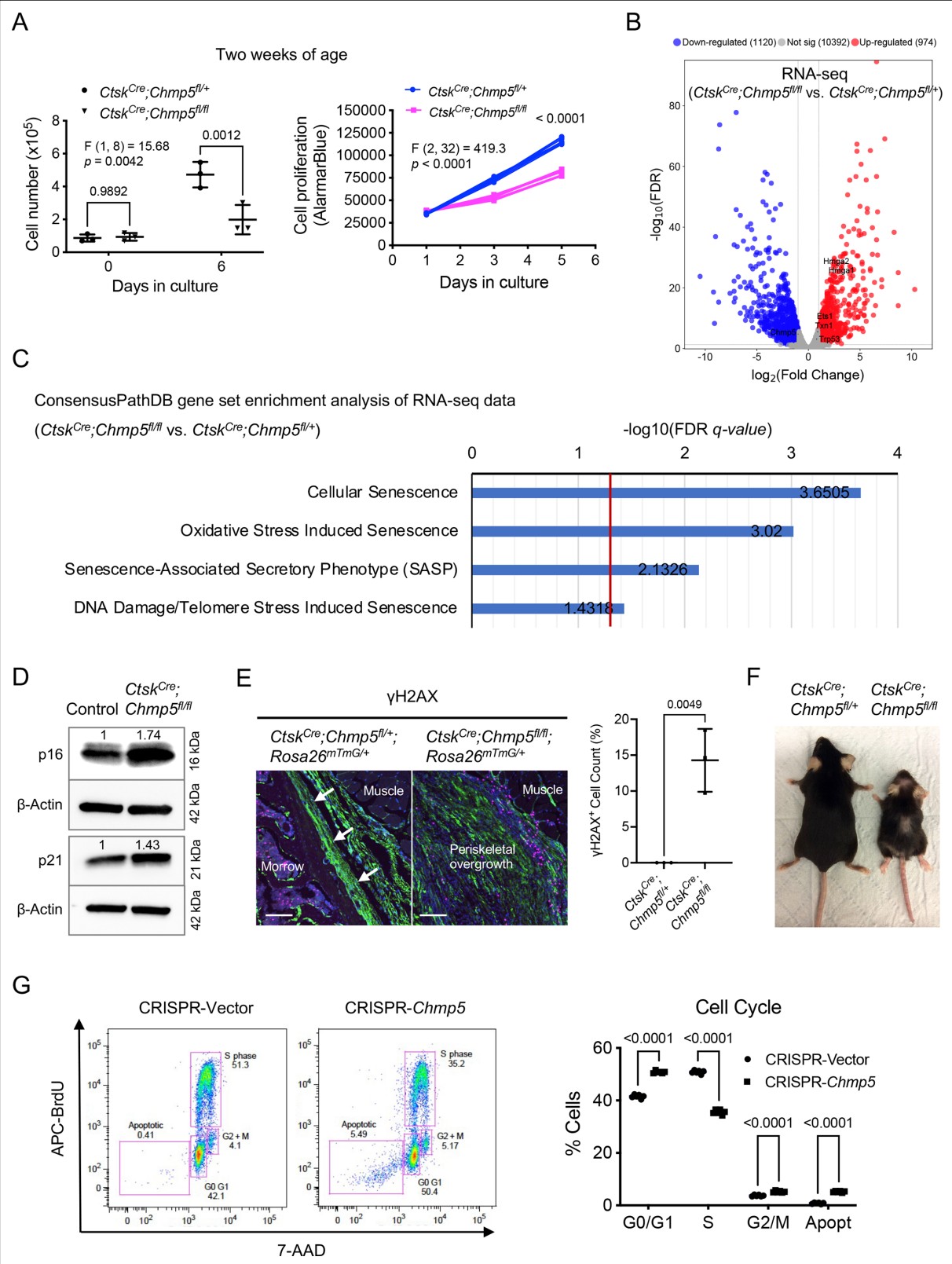

**Figure 3.** Charged multivesicular body protein 5 (CHMP5) controls skeletal progenitor cell senescence. (**A**) Cell number counting and AlamarBlue assay determining cell proliferation in *Ctsk^Cre^;Chmp5^fl/fl^* and *Ctsk^Cre^;Chmp5^fl/+^* periskeletal progenitors from 2-wk-old animals. n=3 replicates per group per time point; experiments repeated three times using cells from three different mice. (**B**) Volcano plot of RNA-seq data showing differentially expressed genes in *Ctsk^Cre^;Chmp5^fl/fl^* vs. *Ctsk^Cre^;Chmp5^fl/+^* periskeletal progenitors. n=3 per group with cells from three different animals. (**C**) ConsensusPathDB

*Figure 3 continued on next page*

*Figure 3 continued*

gene set enrichment analysis of RNA-seq data showing positive enrichment of genes in multiple molecular pathways related to cell senescence (Reactome) in *Ctsk^Cre^;Chmp5^fl/fl^* relative to *Ctsk^Cre^;Chmp5^fl/+^* periskeletal progenitors. (**D**) Western blot determining the expression of p16 and p21 proteins in *Ctsk^Cre^;Chmp5^fl/fl^* compared to wild-type (control) periskeletal progenitors. The numbers above lanes indicate the intensity of the p16 or p21 band relative to that of Control cells after normalization by β-Actin. Results repeated three times with cells from 3 different animals. (**E**) Immunostaining of γH2AX and quantification of γH2AX^+^;GFP^+^ cells in the periosteum and periskeletal overgrowth in *Ctsk^Cre^;Chmp5^fl/+^;Rosa26^mTmG/+^* and *Ctsk^Cre^;Chmp5^fl/fl^;Rosa26^mTmG/+^* mice, respectively. n=3 with tissues from three different animals per group. Scale bars, 100 μm; arrows indicating the periosteum. (**F**) Gross image demonstrating aging-related phenotypes in *Ctsk^Cre^;Chmp5^fl/fl^* compared to *Ctsk^Cre^;Chmp5^fl/+^* mice at 1 y of age. Images are representative of five animals per group. (**G**) Cell cycle analysis in ATDC5 cells with or without *Chmp5* deletion. n=3 replicates per group, results repeated twice. All data are mean ± s.d.; two-way ANOVA followed by multiple comparisons or two-tailed unpaired Student's *t*-test (**E**).

The online version of this article includes the following source data and figure supplement(s) for figure 3:

**Source data 1.** Data of cell proliferation, γH2AX^+^ cell counts, and cell cycle.

**Source data 2.** Original files for western blot analysis displayed in *Figure 3D*.

**Source data 3.** PDF file containing original western blots for *Figure 3D*, indicating the relevant bands.

**Figure supplement 1.** Cell apoptosis is not the main cause of the decreased proliferative rate in charged multivesicular body protein 5 (*Chmp5*)-deficient skeletal progenitors.

**Figure supplement 1—source data 1.** Data on cell apoptosis, gene expression, and cell numbers.

**Figure supplement 1—source data 2.** Original files for western blot analysis displayed in *Figure 3—figure supplement 1E*.

**Figure supplement 1—source data 3.** PDF file containing original western blots for *Figure 3—figure supplement 1E*, indicating the relevant bands.

(*Figure 3*), and the GFP^-^*Ctsk^Cre^;Chmp5^fl/fl^* periskeletal progenitors showed an increased cell proliferation rate (*Figure 4F* and *Figure 4—figure supplement 1C*), these Ki-67^+^ cells in *Figure 4—figure supplement 1E* should represent GFP^-^skeletal progenitors in the periskeletal lesion. Together, these results demonstrate that *Chmp5*-deficient skeletal progenitors could promote the proliferation and osteogenesis of surrounding wild-type skeletal progenitors.

Next, we performed a secretome analysis to characterize secretory profiles of *Ctsk^Cre^;Chmp5^fl/fl^* vs. wild-type periskeletal progenitors. Fifteen proteins were found to increase and five proteins to decrease in the cell supernatant of *Ctsk^Cre^;Chmp5^fl/fl^* periskeletal progenitors (*Figure 4H* and *Figure 4—figure supplement 1F*). In particular, all 15 upregulated proteins, including NCAM1, CSPG4, SDF4, NME2, TAGLN, OLFML2B, PXDN, COL4A1, FSTL1, TPI1, MFGE8, COL1A2, COL1A1, HSPG2, and COL3A1, have been identified in the osteoblastic cell secretome or in the regulation of osteoblast differentiation or functions (*Romanello et al., 2014*; *Davies et al., 2019*). Western blot verified upregulation of two of these proteins, COL1A1 and TAGLN, in the cell supernatant of *Ctsk^Cre^;Chmp5^fl/fl^* vs. wild-type periskeletal progenitors (*Figure 4I*). Notably, the secretome analysis did not detect common SASP factors, such as cytokines and chemokines, in the secretory profile of *Ctsk^Cre^;Chmp5^fl/fl^* periskeletal progenitors, probably due to their small molecular weights and the technical limitations of the mass-spec analysis. Taken together, these results demonstrate a secretory phenotype and paracrine actions of *Chmp5*-deficient skeletal progenitors. However, factors that mediate the paracrine actions of *Chmp5*-deficient periskeletal progenitors remain to be further clarified in future studies.

## Senolytic treatment mitigates musculoskeletal pathologies in *Chmp5* conditional knockout mice

To affirm the role of cell senescence of osteogenic cells in contributing to bone overgrowth in *Ctsk^Cre^;Chmp5^fl/fl^* and *Dmp1^Cre^;Chmp5^fl/fl^* mice, we treated *Chmp5*-deficient periskeletal progenitors and animals with the senolytic drugs quercetin and dasatinib (Q+D), which have been widely used to eliminate senescent cells under various physiological and pathological conditions (*Zhu et al., 2015*; *Hickson et al., 2019*), and assessed the therapeutic efficacy. First, in vitro Q+D treatment showed that *Ctsk^Cre^;Chmp5^fl/fl^* periskeletal progenitors were more sensitive to senolytic drugs for cell apoptosis compared to control cells (*Figure 5A*). *Ctsk^Cre^;Chmp5^fl/fl^* animals were then treated with Q+D from the first 2 wk after birth for 7 wk and showed significantly improved periskeletal bone overgrowth and animal motility compared to *Chmp5^fl/fl^* control animals (*Figure 5B*, *Figure 5—figure supplement 1*). Similarly, treatment of *Dmp1^Cre^;Chmp5^fl/fl^* mice with Q+D also improved musculoskeletal manifestations, including hindlimb abduction, the whole femur and cortical bone thickness, and partially restored skeletal muscle functions (*Figure 5C–E*). An improvement in animal motility was also found

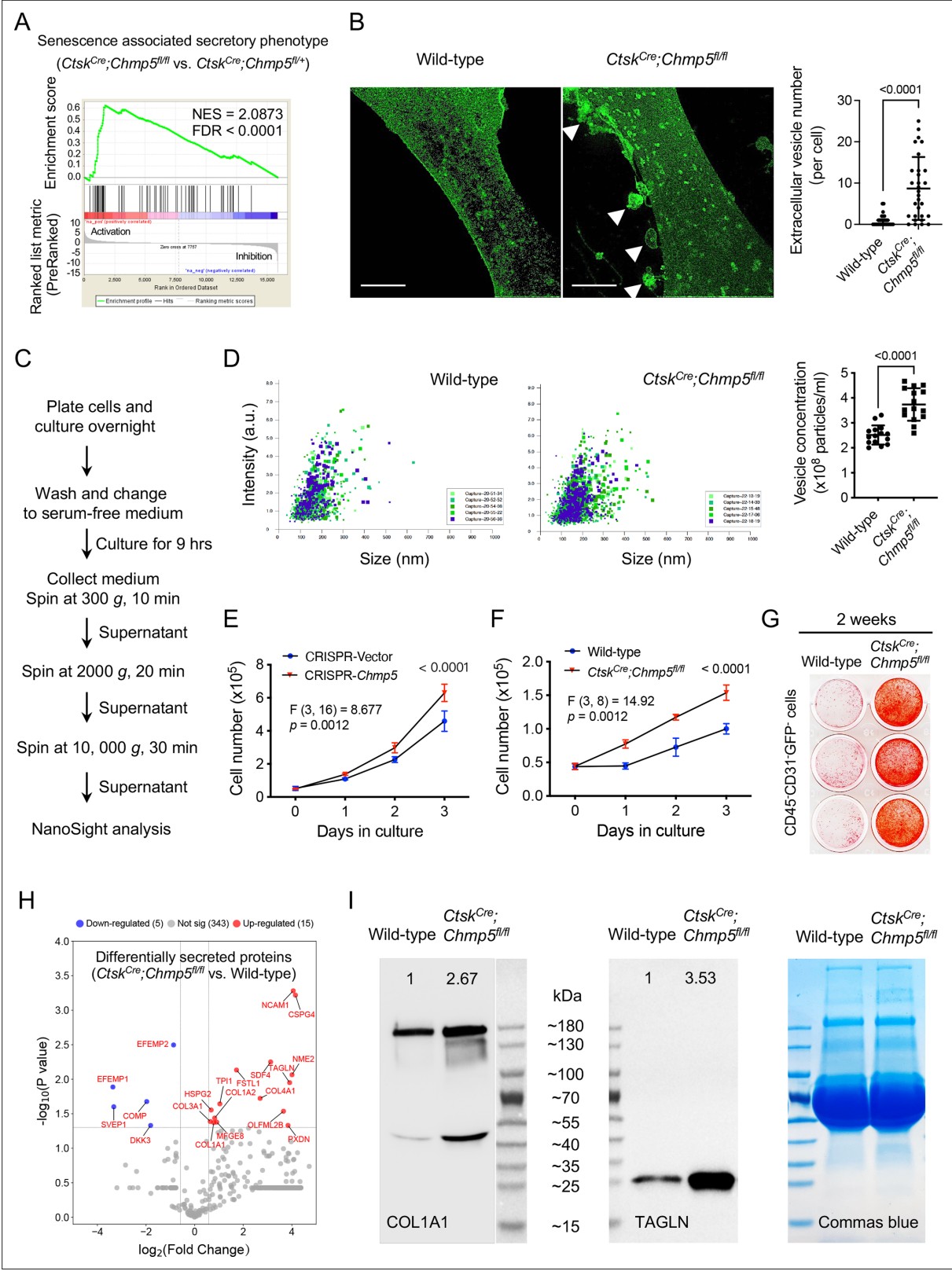

**Figure 4.** Secretory phenotype of charged multivesicular body protein 5 (*Chmp5*)-deficient skeletal progenitors. (**A**) Gene set enrichment analysis (GSEA) of RNA-seq data showing positive enrichment of genes associated with the molecular pathway of senescence-associated secretory phenotype (Reactome database) in *Ctsk^Cre^;Chmp5^fl/fl^* relative to *Ctsk^Cre^;Chmp5^fl/+^* periskeletal progenitors. (**B**) Confocal images showing and quantifying extracellular vesicles (arrowheads) around *Ctsk^Cre^;Chmp5^fl/fl^* periskeletal progenitors. Images are representative of 30 cells from three animals per group. Scale bars,

*Figure 4 continued on next page*

*Figure 4 continued*

10 µm. (**C** and **D**) Nanoparticle tracking analysis of extracellular vesicles in culture medium of *Ctsk^Cre^;Chmp5^fl/fl^* and wild-type periskeletal progenitors. Data pooled from three replicates per group, five reads for each; repeated twice using cells from two mice per group. (**E**) Cell number counting determining cell proliferation in ATDC5 cells treated with culture supernatant from *Chmp5*-deficient or *Chmp5*-sufficient cells. n=3 per group per time point, repeated twice. (**F** and **G**) Cell number counting and alizarin red staining examining cell proliferation (**F**) and osteogenesis (**G**) in neighboring CD45^-^;CD31^-^;GFP^-^ progenitors sorted from periskeletal tissues of *Ctsk^Cre^;Chmp5^fl/fl^;Rosa26^mTmG/+^* or *Ctsk^Cre^;Rosa26^mTmG/+^* mice. n=3 replicates per group per time point, repeated three times using cells from three different animals. (**H**) Volcano plot showing differentially secreted protein in supernatants of *Ctsk^Cre^;Chmp5^fl/fl^* vs. wild-type periskeletal progenitors analyzed by Nano LC-MS/MS. n=3 with cells from three different animals per group. (**I**) Western blot verifying the increase in COL1A1 and TAGLN proteins in supernatants of *Ctsk^Cre^;Chmp5^fl/fl^* relative to wild-type periskeletal progenitors. Commas blue stain as the loading control. Results repeated three times using cells from three different mice in each group. All data are mean ± s.d.; two-tailed unpaired Student's *t*-test for comparison of two groups or two-way ANOVA followed by multiple comparisons.

The online version of this article includes the following source data and figure supplement(s) for figure 4:

**Source data 1.** Data of extracellular vesicle number, concentration, and cell numbers.

**Source data 2.** Original files for western blot analysis displayed in *Figure 4I*.

**Source data 3.** PDF file containing original western blots for *Figure 4I*, indicating the relevant bands.

**Figure supplement 1.** Secretory phenotype of charged multivesicular body protein 5 (*Chmp5*)-deficient skeletal progenitors.

**Figure supplement 1—source data 1.** Data of extracellular vesicle distribution, Alizarin red S quantification, Alamar blue activity, and Ki-67^+^ cell counts.

*with* Q+D *treatment in adult mice.* There were no apparent abnormalities in *Chmp5^fl/fl^* control animals treated with the same dose of Q+D in parallel.

Furthermore, to verify the efficacy of senolytic drugs in treating the periskeletal bone overgrowth in *Ctsk^Cre^;Chmp5^fl/fl^* mice, we used another senolytic drug Navitoclax (ABT-263), which is a BCL-2 family inhibitor and specifically induces apoptosis of senescent cells (**Tse et al., 2008**; **Chang et al., 2016**). Micro-CT analysis demonstrated that ABT-263 could also relieve periskeletal bone overgrowth in *Ctsk^Cre^;Chmp5^fl/fl^* mice (**Figure 5F**). Together, these results demonstrate that the elimination of senescent cells using senolytic drugs is effective in alleviating musculoskeletal pathologies in *Ctsk^Cre^;Chmp5^fl/fl^* and *Dmp1^Cre^;Chmp5^fl/fl^* animals, confirming that osteogenic cell senescence is responsible for musculoskeletal abnormalities in *Ctsk^Cre^;Chmp5^fl/fl^* and *Dmp1^Cre^;Chmp5^fl/fl^* mice.

## CHMP5 is essential for endolysosomal functions and maintaining VPS4A protein in skeletal progenitors

Next, we wondered whether *Chmp5* deficiency in skeletal progenitors affects the endocytic pathway. Notably, approximately 45% of *Ctsk^Cre^;Chmp5^fl/fl^* periskeletal progenitors contained enlarged GFP^+^ vesicles, which were rarely found in wild-type *Ctsk^+^* periskeletal progenitors (**Figure 6A**), suggesting a possible issue with the endocytic pathway in *Chmp5*-deleted skeletal progenitors. Next, we used molecular markers for different components and the function of the endocytic pathway to elucidate the influence of *Chmp5* deficiency on the endolysosomal system in skeletal progenitors (**Figure 6B**).

The LysoTracker Red DND-99 trace and the immunostaining of lysosome-associated membrane protein 1 (LAMP1) showed positive staining in many of the enlarged vesicles in *Ctsk^Cre^;Chmp5^fl/fl^* periskeletal progenitors (**Figure 6C** and **Figure 6—figure supplement 1A, B**). The quantification of the number and average size of LAMP1-stained vesicles were further performed in ATDC5 cells with or without *Chmp5* deletion (**Figure 6D**). Next, the number and size of the vesicles stained for the late endosome marker RAB7 also increased markedly in *Chmp5*-deficient relative to wild-type skeletal progenitors (**Figure 6E and F**). Meanwhile, many of these accumulated vesicles were double positive for LAMP1 and RAB7 (**Figure 6—figure supplement 1C, D**), suggesting that they are terminal compartments in the endocytic pathway. Accordingly, the transmission electron microscopy demonstrated a significant accumulation of MVB-like and lysosome-like structures with different electron density in *Ctsk^Cre^;Chmp5^fl/fl^* periskeletal progenitors (**Figure 6G**).

To examine the function of the endocytic pathway, we used a pH-sensitive fluorescent EGF conjugate and traced the degradation of the internalized EGF receptor in *Ctsk^Cre^;Chmp5^fl/fl^* vs. wild-type periskeletal progenitors. *Chmp5* deficiency significantly delayed the degradation of the EGF conjugate in *Ctsk^Cre^;Chmp5^fl/fl^* periskeletal progenitors (**Figure 6H**), indicating altered function of the endocytic pathway. It should be noted that the early endosomes of EEA1^+^ also increased slightly in approximately 10% of *Ctsk^Cre^;Chmp5^fl/fl^* periskeletal progenitors, while the recycling endosomes of RAB11^+^

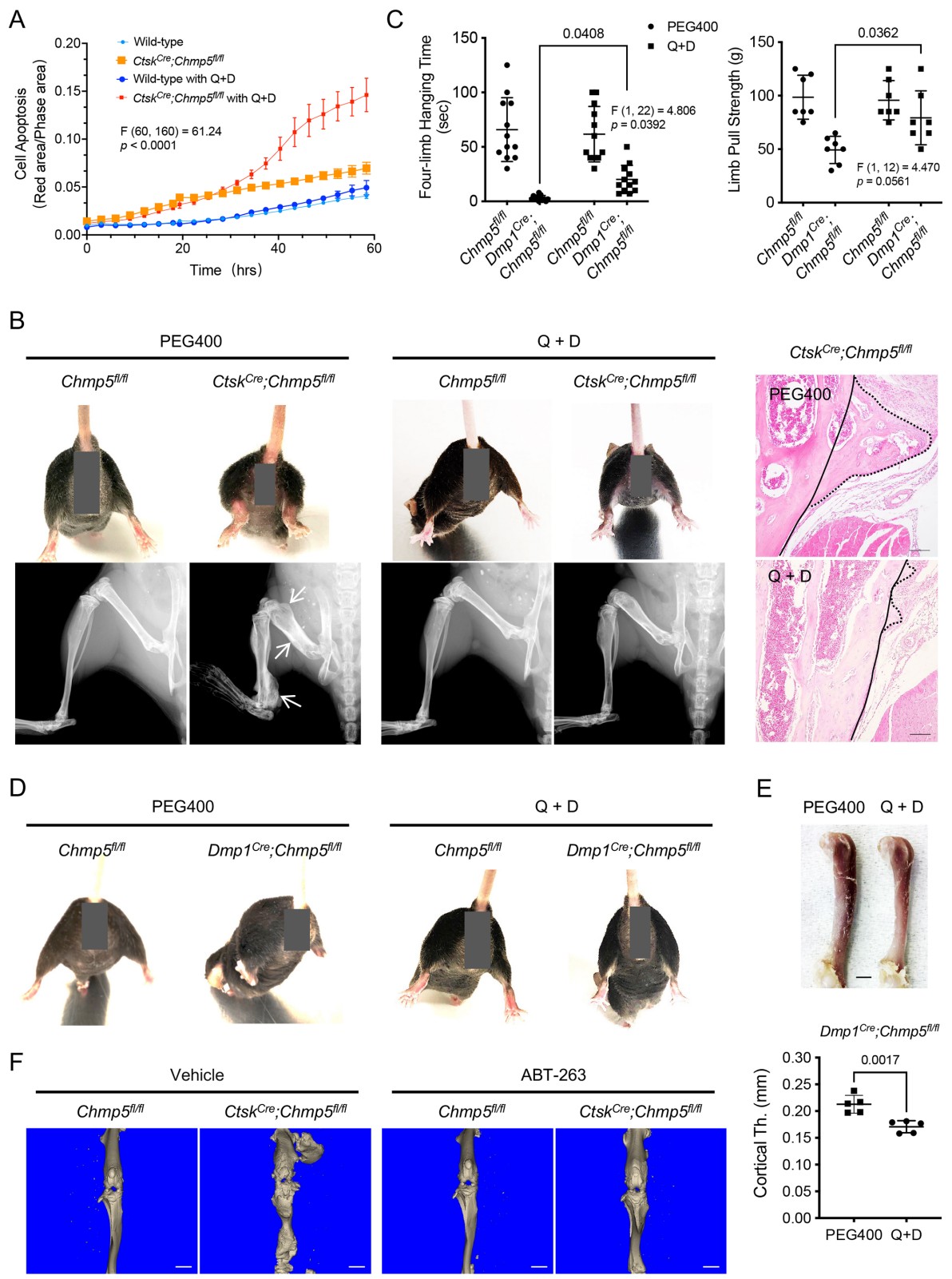

**Figure 5.** Senolytic treatment mitigates musculoskeletal pathologies in charged multivesicular body protein 5 (*Chmp5*) conditional knockout mice. (**A**) Incucyte live-cell apoptosis analysis of *Ctsk^Cre^;Chmp5^fl/fl^* and wild-type periskeletal progenitors treated with 50 μM Quercetin and 500 nM dasatinib (Q+D) and labeled by Annexin V. n=6 for each group, experiment repeated twice using cells from different animals. (**B**) Gross images, radiography, and histology (H&E) respectively demonstrating hindlimb abduction and periskeletal bone overgrowth (arrows or dot-line) in *Ctsk^Cre^;Chmp5^fl/fl^* in comparison

*Figure 5 continued on next page*

*Figure 5 continued*

with *Chmp5^fl/fl* mice after treatment with Q+D or vehicle PEG400 weekly for 7 wk. n=8–10 mice per group. Continuous lines in H&E images indicating an approximate edge of the cortical bone and dot lines indicating the edge of periskeletal overgrowing bones. Scale bars, 100 µm. (**C**) Four-limb hanging time and forelimb pull strength in *Dmp1^Cre;Chmp5^fl/fl* and *Chmp5^fl/fl* mice after treatment with Q+D or the vehicle PEG400 weekly for 16 wk. n=12 animals per group for hanging time test pooled from both genders; n=7 male mice per group for the forelimb pull strength test, similar changes found in both genders. (**D**) Gross images showing hindlimb abduction in *Dmp1^Cre;Chmp5^fl/fl* compared to *Chmp5^fl/fl* mice after treatment with Q+D or the vehicle PEG400. n=12 animals per group. (**E**) Gross image and Micro-CT analysis demonstrating femur and cortical bone thickness in *Dmp1^Cre;Chmp5^fl/fl* mice after treatment with Q+D or the vehicle PEG400. n=5 male mice per group, similar changes found in both genders. Scale bar, 1 mm. (**F**) Micro-CT images showing periskeletal bone overgrowth in *Ctsk^Cre;Chmp5^fl/fl* in comparison with *Chmp5^fl/fl* mice after treatment with ABT-263 or vehicle weekly for 8 wk. n=4 mice per group, Scale bars, 2 mm. All data are mean ± s.d.; two-way ANOVA followed by multiple comparisons or two-tailed unpaired Student's *t*-test for comparison of two groups.

The online version of this article includes the following source data and figure supplement(s) for figure 5:

**Source data 1.** Data on cell apoptosis, skeletal muscle functions, and cortical bone thickness after senolytic treatments.

**Figure supplement 1.** Additional x-ray images of *Ctsk^Cre;Chmp5^fl/fl* mice after treatment with Q+D or the vehicle PEG400 for 7 wk.

and the TGN38$^+$ trans-Golgi network were relatively normal (*Figure 6—figure supplement 1E*). These results demonstrate that CHMP5 deficiency in skeletal progenitors causes the accumulation of late endosomes and lysosomes and disrupts the function of the endolysosomal pathway.

To dissect the molecular mechanism of CHMP5 that regulates the endolysosomal pathway, we performed a proteomic analysis in *Ctsk^Cre;Chmp5^fl/fl* and wild-type periskeletal progenitors. Bioinformatic analyses of proteomic data showed that VPS4A is one of the most significantly decreased proteins in *Ctsk^Cre;Chmp5^fl/fl* compared to wild-type periskeletal progenitors (*Figure 6I*). The decrease in the VPS4A protein in *Ctsk^Cre;Chmp5^fl/fl* periskeletal progenitors was verified by Western blot analysis (*Figure 6J*), while the *Vps4a* mRNA did not show a significant change in the RNA-seq analysis (*Figure 6—figure supplement 1F*). On the other hand, when CHMP5 was overexpressed by transfection of a CHMP5 vector into HEK-293T cells, the level of VPS4A protein increased significantly and *VPS4A* mRNA was not significantly affected (*Figure 6—figure supplement 1G, H*). These results indicate that CHMP5 is essential in maintaining the VPS4A protein. Since VPS4A is a critical ATPase mediating the function of the ESCRT-III protein complex, the reduction of VPS4A protein could be responsible for endolysosomal dysfunction in *Chmp5*-deleted skeletal progenitors.

## Mitochondrial dysfunction is responsible for cell senescence in *Chmp5*-deleted skeletal progenitors

A remaining question is how *Chmp5* deficiency and endolysosomal dysfunction cause skeletal cell senescence. Impairment of lysosomal functions could hinder the degradation of damaged mitochondria, resulting in the accumulation of reactive oxygen species (ROS) in mitochondria (*Martini and Passos, 2023*; *Park et al., 2018*). Indeed, the GSEA of RNA-seq data showed significant enrichment of genes associated with the molecular pathway of oxidative stress-induced cell senescence in *Ctsk^Cre;Chmp5^fl/fl* vs. *Ctsk^Cre;Chmp5^fl/+* periskeletal progenitors (*Figure 7A*). Accordingly, the level of mitochondrial ROS and the abundance of mitochondria increased markedly in *Chmp5*-deficient compared to wild-type skeletal progenitors, as shown by CellRox, tetramethylrhodamine methyl ester (TMRE), and MitoTracker fluorescence stain, and expression levels of mitochondrial inner membrane proteins NDUF88, SDHB, and ATP5A (*Figure 7B–D* and *Figure 7—figure supplement 1A, B*).

Mitochondrial functions were also compromised in *Chmp5*-deficient skeletal progenitors as shown by decreased mitochondrial respiratory capacity, increased extracellular acidification rate, and reduced tolerance to galactose (*Figure 7E and F*). Furthermore, mitochondrial dysfunction and DNA damage response usually form a feedback loop to drive senescent cell phenotypes (*Gorgoulis et al., 2019*; *Korolchuk et al., 2017*; *Birch and Passos, 2017*). The GSEA of the RNA-seq data also showed enrichment of genes associated with DNA damage-induced cell senescence in *Ctsk^Cre;Chmp5^fl/fl* vs. *Ctsk^Cre;Chmp5^fl/+* skeletal progenitors (*Figure 7—figure supplement 1C*). Upregulation of two critical DNA damage-responsive genes *H2afx* and *Trp53* was verified by qPCR in *Ctsk^Cre;Chmp5^fl/fl* relative to *Ctsk^Cre;Chmp5^fl/+* skeletal progenitors (*Figure 7—figure supplement 1D*). Importantly, N-Acetylcysteine (NAC) antioxidant treatment could reverse the reduced activity of cell proliferation in *Chmp5*-deficient skeletal progenitors (*Figure 7G*). Therefore, the accumulation of dysfunctional mitochondria and mitochondrial ROS is responsible for cell senescence in *Chmp5*-deficient skeletal progenitors.

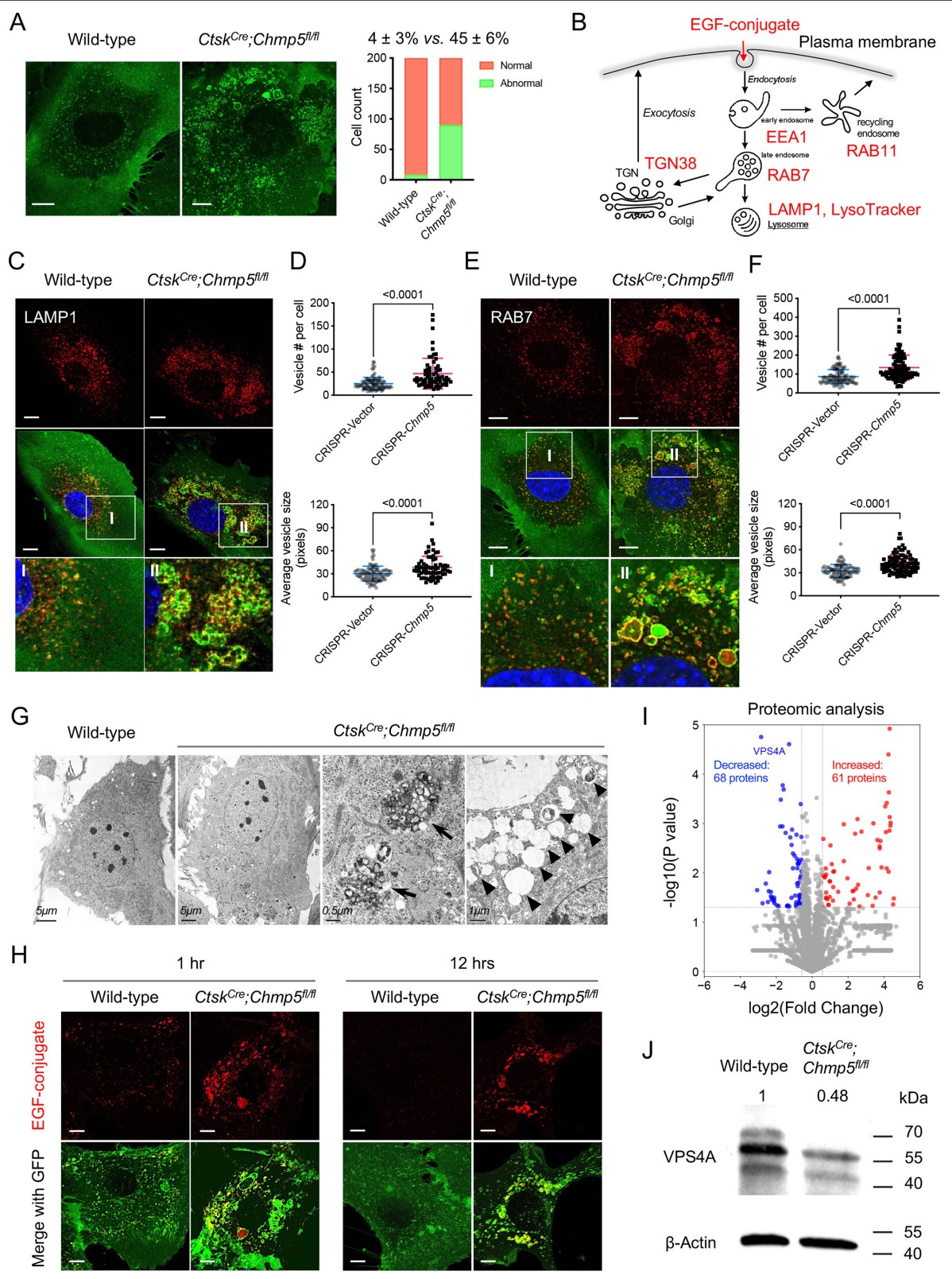

**Figure 6.** Charged multivesicular body protein 5 (CHMP5) is essential for endolysosomal functions and maintaining VPS4A protein in skeletal progenitors. (**A**) Representative confocal images demonstrating abnormally enlarged vesicles in cultured *Ctsk^Cre^;Chmp5^fl/fl^* relative to wild-type periskeletal progenitors. Abnormal cells identified by containing enlarged GFP⁺ vesicles. n=200 cells from three mice per group. (**B**) Schematic showing molecular markers utilized to analyze the endocytic pathway. (**C**) Representative confocal images showing LAMP1 immunostaining in *Ctsk^Cre^;Chmp5^fl/*

*Figure 6 continued on next page*

*Figure 6 continued*

*fl* and wild-type periskeletal progenitors. n=20 cells per group. (**D**) Quantification of LAMP1+ vesicles in ATDC5 cells with or without *Chmp5* depletion. n=93, 66 cells, respectively. (**E**) Representative confocal images showing RAB7 immunostaining in *Ctsk^Cre^;Chmp5^fl/fl^* and wild-type periskeletal progenitors. n=15 cells per genotype. (**F**) Quantification of RAB7+ vesicles in ATDC5 cells with or without *Chmp5* depletion. n=94, 92 cells, respectively. (**G**) Transmission electron microscopy showing the accumulation of multivesicular body-like structures (arrows) and lysosome-like structures (arrowheads) in *Ctsk^Cre^;Chmp5^fl/fl^* relative to wild-type periskeletal progenitors. n=30 cells per group. (**H**) Confocal live cell images demonstrating delayed degradation of the EGF conjugate in *Ctsk^Cre^;Chmp5^fl/fl^* vs. wild-type periskeletal progenitors. n=10 cells each group per time point. (**I**) Volcano plot of proteomic analysis showing differentially expressed proteins in *Ctsk^Cre^;Chmp5^fl/fl^* vs. wild-type periskeletal progenitors. n=3 with cells from three different animals. (**J**) Western blot verifying the decrease in VPS4A protein in *Ctsk^Cre^;Chmp5^fl/fl^* vs. wild-type periskeletal progenitors. The numbers above lanes indicate the intensity of the VPS4A band relative to that of wild-type cells after normalization by β-Actin. Results repeated three times with cells from three different mice in each group. All data are mean ± s.d.; Mann-Whitney test for the comparison of the vesicle number and two-tailed unpaired Student's *t*-test for the comparison of the vesicle size in (**D**) and (**F**). Scale bars, 10 μm except (**G**) as indicated.

The online version of this article includes the following source data and figure supplement(s) for figure 6:

**Source data 1.** Data of LAMP1+ or RAB7+ vesicle numbers and average sizes.

**Source data 2.** Original files for western blot analysis displayed in *Figure 6J*.

**Source data 3.** PDF file containing original western blots for *Figure 6J*, indicating the relevant bands.

**Figure supplement 1.** Charged multivesicular body protein 5 (CHMP5) is essential for endolysosomal functions and maintaining VPS4A protein in skeletal progenitors.

**Figure supplement 1—source data 1.** Data of LysoTracker fluorescence intensity, colocalization of LAMP1 and RAB7, and gene expression.

**Figure supplement 1—source data 2.** Original files for western blot analysis displayed in *Figure 6—figure supplement 1G*.

**Figure supplement 1—source data 3.** PDF file containing original western blots for *Figure 6—figure supplement 1G*, indicating the relevant bands.

## Discussion

In this study, our results reveal the role and mechanism of CHMP5 in the regulation of cell senescence and bone formation in osteogenic cells. In normal skeletal progenitor cells, CHMP5 is essential in maintaining the VPS4A protein, a critical AAA ATPase for the functions of the ESCRT-III protein complex. The deficiency of CHMP5 decreases the VSP4A protein and causes the accumulation of dysfunctional late endosomes (MVB), lysosomes, and mitochondria, which induce skeletal cell senescence and result in increased bone formation through a combination of cell-autonomous and paracrine mechanisms (*Figure 7H*). Strikingly, the elimination of senescent cells using senolytic drugs is effective in mitigating abnormal bone formation.

The role of CHMP5 in cell senescence has not been reported. As the mechanisms and functions of cell senescence could be highly heterogeneous depending on inducers, tissue and cell contexts, and other factors such as 'time' (*Paramos-de-Carvalho et al., 2021*; *Wiley and Campisi, 2021*; *Martini and Passos, 2023*), there is currently no universal molecular marker to define all types of cell senescence. In this study, *Chmp5*-deficient skeletal progenitors show multiple canonical features of senescent cells, including increased p16 and p21 proteins and γH2Ax+ cells, cell cycle arrest, elevated secretory phenotype, accumulation and dysfunction of lysosomes and mitochondria. Also, *Ctsk^Cre^;Chmp5^fl/fl^* and *Dmp1^Cre^;Chmp5^fl/fl^* mice show dramatic aging-related phenotypes, including hair loss, joint stiffness/contracture, decreased bone strength, or gradual loss of muscle mass and functions. Importantly, senolytic drugs markedly improve musculoskeletal abnormalities and increase animal motility in *Ctsk^Cre^;Chmp5^fl/fl^* and *Dmp1^Cre^;Chmp5^fl/fl^* mice. These results reveal the role of CHMP5 in controlling the senescence of skeletal cells.

CHMP5 has been reported to regulate VPS4 activity, which catalyzes the disassembly of the ESCRT-III protein complex from late endosomes/MVBs, by binding directly to the VPS4 protein or other protein partners such as LIP5 and Brox (*Vild et al., 2015*; *Yang et al., 2012*). However, it remains unknown how CHMP5 regulates VPS4 activity. Our results in this study show that CHMP5 is necessary to maintain the VPS4A protein but does not affect the level of VPS4A mRNA, suggesting that CHMP5 regulates VPS4 activity by maintaining the level of the VPS4A protein. However, the mechanism of CHMP5 in the regulation of the VPS4A protein has not yet been studied. Since CHMP5 can recruit the deubiquitinating enzyme USP15 to stabilize IκBα in osteoclasts by suppressing ubiquitination-mediated proteasomal degradation (*Greenblatt et al., 2015*), it is also possible that CHMP5 stabilizes the VPS4A protein by recruiting deubiquitinating enzymes and regulating the ubiquitination of VPS4A, which needs to be clarified in future studies. Notably, mutations in the *VPS4A* gene in humans can

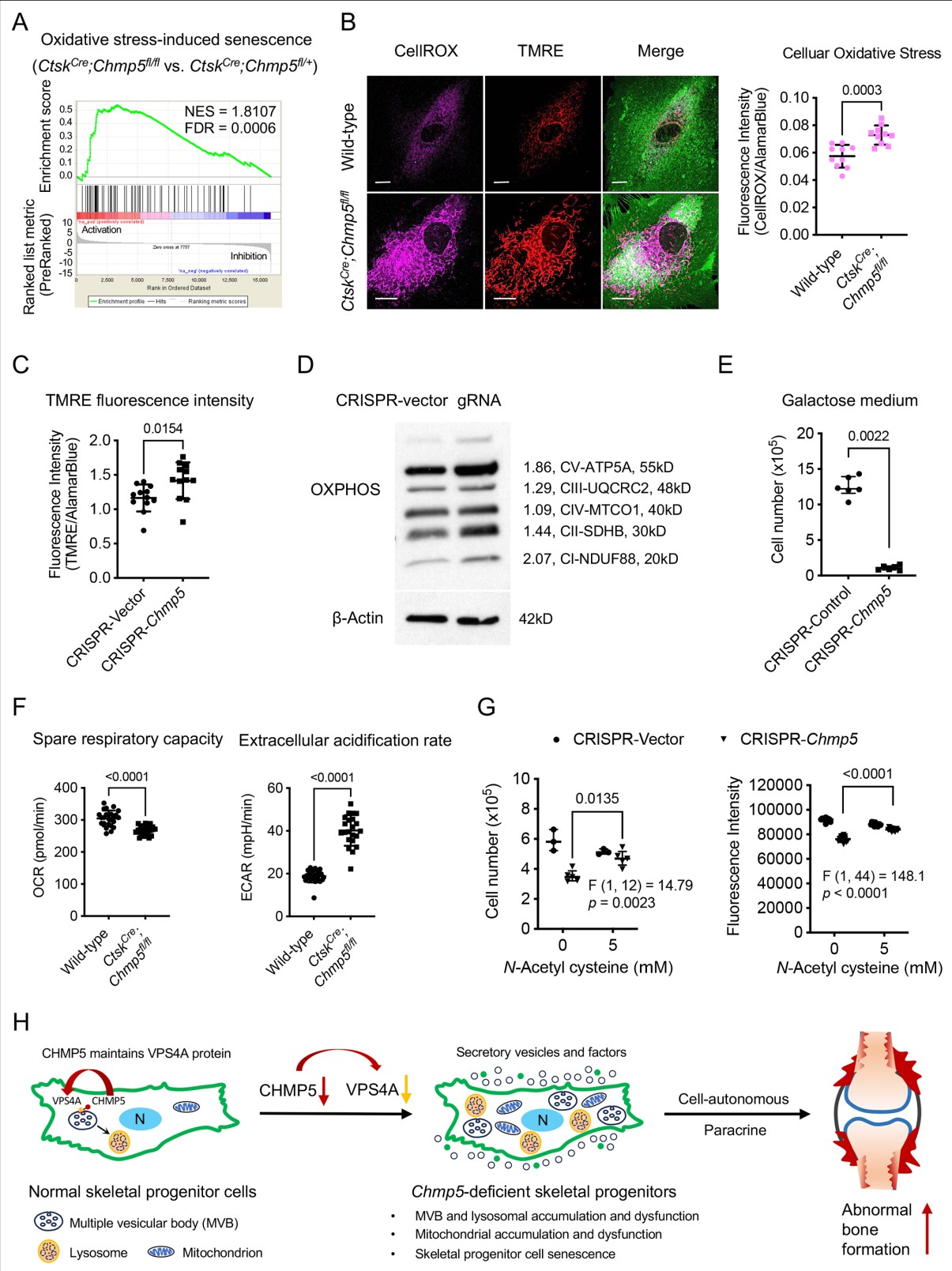

**Figure 7.** Mitochondrial dysfunction is responsible for cell senescence in charged multivesicular body protein 5 (*Chmp5*)-deleted skeletal progenitors. (**A**) Gene set enrichment analysis (GSEA) of RNA-seq data reporting positive enrichment of genes associated with the molecular pathway of oxidative stress-induced senescence in *Ctsk^Cre^;Chmp5^fl/fl^* relative to *Ctsk^Cre^;Chmp5^fl/+^* periskeletal progenitors. (**B**) Confocal fluorescence images showing intracellular ROS (CellROX Deep Red) and mitochondria (TMRE) in *Ctsk^Cre^;Chmp5^fl/fl^* vs. wild-type periskeletal progenitors. Images are representative of

*Figure 7 continued on next page*

*Figure 7 continued*

30 cells per group. Scale bars, 15 μm. The graph on the right shows the quantification of the fluorescence intensity of CellROX Deep Red. n=10 each group, results repeated twice. (**C**) Quantification of TMRE fluorescence intensity in *Chmp5*-sufficient and *Chmp5*-deficient ATDC5 cells. n=12 for each group, repeated three times. (**D**) Western blotting determines the expression of mitochondrial OXPHOS proteins NDUF88, SDHB, MT-CO1, UQCRC2, and ATF5A in *Chmp5*-deficient compared to *Chmp5*-sufficient ATDC5 cells. Experiment was repeated twice. (**E**) Cell number counting to determine cell proliferation in galactose medium. n=6 replicates per group, repeated three times. (**F**) Seahorse mitochondrial stress test showing mitochondrial respiratory capacity and extracellular acidification rate in *Ctsk^Cre;Chmp5^fl/fl* relative to wild-type periskeletal progenitors. Data pooled from cells of three mice, eight replicates for each sample. (**G**) Cell number counting and AlamarBlue assay determining cell proliferation in ATDC5 cells with or without *Chmp5* depletion after treatment with the antioxidant *N*-Acetyl cysteine. n=3 or 12 each group for cell number counting or AlamarBlue assay, respectively; results repeated three times. (**H**) Schematic showing the function of CHMP5 in maintaining VPS4A protein and endolysosomal homeostasis and restricting cell senescence and osteogenesis in skeletal progenitors. All data are mean ± s.d.; two-tailed unpaired Student's *t*-test, except the Mann-Whitney test in panel (**E**).

The online version of this article includes the following source data and figure supplement(s) for figure 7:

**Source data 1.** Data of CellROX and TMRE fluorescence intensity, mitochondrial respiration, and cell proliferation.

**Source data 2.** Original files for western blot analysis displayed in *Figure 7D*.

**Source data 3.** PDF file containing original western blots for *Figure 7D*, indicating the relevant bands.

**Figure supplement 1.** Mitochondrial dysfunction and activation of DNA damage-induced senescence in charged multivesicular body protein 5 (*Chmp5*)-deficient skeletal progenitors.

**Figure supplement 1—source data 1.** For *Figure 7—figure supplement 1D*, Relative gene expression of *H2afx* and *Trp53* ($2^{-\Delta Ct}$).

cause multisystemic diseases, including musculoskeletal abnormalities (*Rodger et al., 2020*) (OMIM: 619273), suggesting that normal expression and function of VPS4A are important for musculoskeletal physiology. The roles of VPS4A in regulating musculoskeletal biology and cell senescence should be further explored in future studies.

The upstream signaling that regulates CHMP5 expression in osteogenic cells is unclear. Previous studies showed that muramyl-dipeptide and lipopolysaccharide upregulate CHMP5 expression in human monocytes (*Salazar et al., 2009*; *Thiébaut et al., 2016*). Whether the mechanism also occurs in osteogenic cells remains to be determined, especially since many molecular and cellular mechanisms are cell lineage/type dependent. In addition, an earlier study showed that *Rank* haploinsufficiency in *Ctsk^Cre;Chmp5^fl/fl* mice (*Ctsk^Cre;Chmp5^fl/fl; Rank^+/−*) could greatly reverse skeletal abnormalities in these animals, including periskeletal overgrowth (*Greenblatt et al., 2015*). Therefore, RANK receptor signaling may function upstream of CHMP5. However, whether RANK is expressed in osteogenic cells needs additional studies to clarify. Otherwise, other epigenetic, transcriptional, translational, or post-translational mechanisms may also play a role in the regulation of CHMP5 expression in osteogenic lineage cells in physiological and pathological contexts.

A previous study using the *Ctsk^Cre;Chmp5^fl/fl* mouse model defined a function of CHMP5 in the regulation of osteoclast differentiation by tuning NF-κB signaling (*Greenblatt et al., 2015*). While the evidence on the role of CHMP5 in osteoclastogenesis is robust in that study, our results in the current study provide compelling evidence on the role of CHMP5 in bone formation in osteogenic cells. Furthermore, the function of CHMP5 in restricting bone formation was confirmed in the *Dmp1^Cre;Chmp5^fl/fl* mouse model and in the MC3T3-E1 cell model (*Figures 1 and 2*). Therefore, it is probable that both osteoclast and osteoblast lineage cells contribute to bone deformation in *Ctsk^Cre;Chmp5^fl/fl* mice. However, the previous study by Greenblatt et al. did not find endolysosomal abnormalities in osteoclasts after *Chmp5* deletion, and in that study, cell senescence and mitochondrial functions were not measured. Also, a previous study by Adoro et al. did not detect endolysosomal abnormalities in *Chmp5*-deficient developmental T cells (*Adoro et al., 2017*). Since both osteoclasts and T cells are of hematopoietic origin, and meanwhile osteogenic cells and MEFs, which show endolysosomal abnormalities after CHMP5 deficiency, are of mesenchymal origin, it turns out that the function of CHMP5 in regulating the endolysosomal pathway could be cell lineage-specific, which remains to be clarified in future studies.

Furthermore, it is unclear whether the effect of senolytic drugs in *Ctsk^Cre;Chmp5^fl/fl* mice involves targeting osteoclasts other than osteogenic cells, as osteoclast senescence has not yet been evaluated. However, the efficacy of Q+D in targeting osteogenic cells, which is the focus of the current study, was confirmed in *Dmp1^Cre;Chmp5^fl/fl* mice (*Figure 5C–E*). Additionally, Q+D caused a higher cell apoptotic ratio in *Ctsk^Cre;Chmp5^fl/fl* compared to wild-type periskeletal progenitors in ex vivo culture (*Figure 5A*),

demonstrating the effectiveness of Q+D in targeting osteogenic cells in the $Ctsk^{Cre}$;$Chmp5^{fl/fl}$ model. Furthermore, an alternative senolytic drug ABT-263 could also ameliorate periskeletal bone overgrowth in $Ctsk^{Cre}$;$Chmp5^{fl/fl}$ mice (*Figure 5F*). Together, these results confirm that osteogenic cell senescence is responsible for the bone overgrowth in $Ctsk^{Cre}$;$Chmp5^{fl/fl}$ and $Dmp1^{Cre}$;$Chmp5^{fl/fl}$ mice, and senolytic treatments are effective in alleviating these skeletal disorders.

Notably, aging is associated with decreased osteogenic capacity in marrow stromal cells, which is related to conditions with low bone mass, such as osteoporosis. Rather, aging is also accompanied by increased ossification or mineralization in musculoskeletal soft tissues, such as tendons and ligaments (*Dai et al., 2024*). In particular, the abnormal periskeletal overgrowth in $Ctsk^{Cre}$;$Chmp5^{fl/fl}$ mice was predominantly mapped to the insertion sites of tendons and ligaments on the bone (*Figure 1A and E*), which is consistent with changes during aging and suggests that mechanical stress at these sites could contribute to the aberrant bone growth. These results suggest that skeletal stem/progenitor cells at different sites of musculoskeletal tissues could demonstrate different, even opposite outcomes in osteogenesis, due to cell senescence.

In addition, there was also a mild increase in apoptotic cells in the population of $Chmp5$-deficient skeletal progenitors (approximately 5%). Since anti-apoptotic treatment with the pan-caspase inhibitor Q-VD-Oph could not reverse the decrease in cell number in $Chmp5$-deficient skeletal progenitors, cell apoptosis may not be the main cause of the decrease in cell proliferation rate in these progenitors. It should be noted that although senescent cells are generally resistant to apoptosis, the gradual accumulation of dysfunctional lysosomes and mitochondria in $Chmp5$-deficient skeletal progenitors could result in an overload of toxic macromolecules and metabolites, which could activate the apoptotic pathway to eliminate severely damaged cells. Otherwise, CHMP5 may regulate cell senescence and apoptosis through different mechanisms.

In summary, we have revealed the role of CHMP5 in the regulation of cell senescence and bone formation in osteogenic cells. This study also confirms the essential function of CHMP5 in the endolysosomal pathway that involves the regulation of the VPS4A protein. Additionally, mitochondrial functions are impaired with CHMP5 deletion, which is responsible for cell senescence. Future studies should explore the mechanism of CHMP5 in the regulation of mitochondrial functions. Since $Ctsk^{Cre}$;$Chmp5^{fl/fl}$ and $Dmp1^{Cre}$;$Chmp5^{fl/fl}$ mice recapitulate many cellular and phenotypic features of musculoskeletal lesions in lysosomal storage disease, these animals could be used as preclinical models to investigate the mechanism and test therapeutic drugs for these intractable disorders.

## Methods

### Key resources table

| Reagent type (species) or resource | Designation | Source or reference | Identifiers | Additional information |
|---|---|---|---|---|
| Genetic reagent (*Mus. musculus*) | Chmp5$^{tm2.1Gho}$; Chmp5$^{fl}$ | PMID:26195726 | RRID:MGI:3629214 | A gift from Dr. Sankor Ghosh |
| Genetic reagent (*Mus. musculus*) | Chmp5-flox | This paper | | Generated by CRISPR/Cas9; Targeting exons 4 and 5 of mouse *Chmp5* gene |
| Genetic reagent (*Mus. musculus*) | Ctsk$^{tm1(cre)Ska}$; Ctsk$^{Cre}$ | PMID:17803905 | RRID:MGI:3764465 | |
| Genetic reagent (*Mus. musculus*) | B6N.FVB-Tg(Dmp1-cre)1Jqfe/BwdJ; Dmp1-Cre | PMID:17384025 | Strain #:023047 RRID:IMSR_JAX:023047 | |
| Genetic reagent (*Mus. musculus*) | B6.Cg-Tg(Prrx1-cre)1Cjt/J; Prrx1$^{Cre}$ | PMID:12112875 | Strain #:005584 RRID:IMSR_JAX:005584 | |
| Genetic reagent (*Mus. musculus*) | B6;SJL-Tg(Col2a1-cre)1Bhr/J; Col2a1-Cre | PMID:10686612 | Strain #:003554 RRID:IMSR_JAX:003554 | |
| Genetic reagent (*Mus. musculus*) | B6.Cg-Tg(Sp7-tTA,tetO-EGFP/cre)1Amc/J; Osx1-GFP::Cre | PMID:16854976 | Strain #:006361 RRID:IMSR_JAX:006361 | |
| Genetic reagent (*Mus. musculus*) | Gt(ROSA)26Sor$^{tm4(ACTB-tdTomato,-EGFP)Luo}$; mTmG | PMID:17868096 | Strain #:007676 RRID:IMSR_JAX:007676 | |
| Cell line (*M. musculus*) | MC3T3-E1 Cell Line (Mouse C57BL/6 calvaria) | ECACC | Cat# 99072810, RRID:CVCL_0409 | Authenticated by osteogenic differentiation; tested negative for mycoplasma |
| Cell line (*M. musculus*) | ATDC5 cells (mouse 129 teratocarcinoma-derived osteochondral progenitors) | Sigma-Aldrich | Cat#:99072806 RRID:CVCL_3894 | Authenticated by chondrogenic differentiation; tested negative for mycoplasma |

*Continued on next page*

*Continued*

| Reagent type (species) or resource | Designation | Source or reference | Identifiers | Additional information |
|---|---|---|---|---|
| Cell line (*Homo sapiens*) | HEK-293T cells | China Center for Type Culture Collection | RRID:CVCL_0063 | Authenticated by STR profiling; tested negative for mycoplasma |
| Transfected construct (mammalian) | lentiCRISPRv2 | Addgene | Cat# 52961 RRID:Addgene_52961 | A gift from Feng Zhang |
| Transfected construct (mammalian) | pMDLg/pRRE | Addgene | Cat# 12251; RRID:Addgene_12251 | A gift from Didier Trono |
| Transfected construct (mammalian) | pRSV-Rev | Addgene | Cat# 12253; RRID:Addgene_12253 | A gift from Didier Trono |
| Transfected construct (mammalian) | pMD2.G | Addgene | Cat# 12259; RRID:Addgene_12259 | A gift from Didier Trono |
| Antibody | APC anti-mouse CD45 (rat monoclonal) | BioLegend | Cat# 147708, RRID:AB_2563540 | FACS (1:200) |
| Antibody | APC anti-mouse CD31 (rat monoclonal) | BioLegend | Cat# 102410, RRID:AB_312905 | FACS (1:200) |
| Antibody | anti-CHMP5 antibody (Rabbit polyclonal) | PMID:26195726 | | WB (1:1000) |
| Antibody | Anti-CHMP5 antibody (mouse monoclonal) | Santa Cruz Biotechnology | Cat# sc-374338 RRID:AB_10989738 | WB (1:1000) |
| Antibody | Anti-p16 INK4A antibody (rabbit monoclonal) | Cell Signaling Technology | Cat# 29271, RRID:AB_3674659 | WB (1:1000) |
| Antibody | Anti-p21 Waf1/Cip1 antibody (rabbit monoclonal) | Cell Signaling Technology | Cat# 37543, RRID:AB_2935811 | WB (1:1000) |
| Antibody | Anti-phospho-Histone H2A.X (Ser139) antibody (rabbit monoclonal) | Cell Signaling Technology | Cat# 9718, RRID:AB_2118009 | IF (1:400) |
| Antibody | Anti-COL1A1 antibody (rabbit monoclonal) | Cell Signaling Technology | Cat# 72026, RRID:AB_2904565 | WB (1:1000) |
| Antibody | Anti-TAGLN antibody (rabbit polyclonal) | Abcam | Cat# ab14106, RRID:AB_443021 | WB (1:1000) |
| Antibody | Anti-Ki67 antibody (rabbit polyclonal) | Abcam | Cat# ab15580, RRID:AB_443209 | IHC (1:200) |
| Antibody | Anti-EEA1 antibody (mouse monoclonal) | BD Biosciences | Cat# 610456, RRID:AB_397829 | IF (1:150) |
| Antibody | Anti-RAB7 antibody (Rabbit monoclonal) | Cell Signaling Technology | Cat# 9367, RRID:AB_1904103 | IF (1:250) |
| Antibody | Anti-RAB11 antibody (Rabbit monoclonal) | Cell Signaling Technology | Cat# 5589, RRID:AB_10693925 | IF (1:250) |
| Antibody | Anti-CD107a antibody (rat monoclonal) | BD Biosciences | Cat# 553792, RRID:AB_2134499 | IF (1:250) |
| Antibody | Anti-TGN38 antibody (rat monoclonal) | R and D Systems | Cat# MAB7944, RRID:AB_2713953 | IF (1:250) |
| Antibody | Anti-VPS4A antibody (mouse monoclonal) | Santa Cruz Biotechnology | Cat# sc-393428, RRID:AB_2773025 | WB (1:1000) |
| Antibody | Total OXPHOS rodent antibody cocktail | Abcam | Cat# ab110413, RRID:AB_2629281 | WB (1:1000) |
| Antibody | Anti-GAPDH antibody (rabbit monoclonal) | Cell Signaling Technology | Cat# 2118, RRID:AB_561053 | WB (1:2000) |
| Antibody | Anti-β-Actin antibody (rabbit monoclonal) | Cell Signaling Technology | Cat# 8457, RRID:AB_10950489 | WB (1:2000) |
| Recombinant DNA reagent | pCMV-SFB-CHMP5-Neo (plasmid) | This paper | | Overexpression of human CHMP5 |
| Recombinant DNA reagent | pCMV-SFB-Neo (plasmid) | This paper | | Empty vector |
| Sequence-based reagent | sgRNAs targeting mouse *Chmp5* exon 1 | This paper | | sgRNA1: GGCTCCGCCACCTAGCTTGA, sgRNA2: GTTT CGCTTTTCCGAAGAAT |
| Peptide, recombinant protein | pHrodo Red Epidermal Growth Factor (EGF) Conjugate | ThermoFisher Scientific | Cat# P35374 | |
| Commercial assay or kit | TUNEL Assay Kit - HRP-DAB | Abcam | Cat# ab206386 | |
| Commercial assay or kit | Annexin V Apoptosis Detection Kit APC | ThermoFisher Scientific | Cat# 88-8007-72 | |

*Continued on next page*

*Continued*

| Reagent type (species) or resource | Designation | Source or reference | Identifiers | Additional information |
|---|---|---|---|---|
| Commercial assay or kit | APC BrdU Flow Kit | BD Bioscience | Cat# 552598 | |
| Commercial assay or kit | Seahorse XF Cell Mito Stress Test Kit | Agilent | Cat# 103015–100 | |
| Chemical compound, drug | InSolution Q-VD-OPh, Non-O-methylated | Millipore | Cat# 551476 | 100 µM |
| Chemical compound, drug | Lysotracker Red DND-99 | ThermoFisher Scientific | Cat# L7528 | 75 nM |
| Chemical compound, drug | MitoTracker Green FM | ThermoFisher Scientific | Cat# M7514 | 200 nM |
| Chemical compound, drug | CellROX Deep Red Reagent | ThermoFisher Scientific | Cat# C10422 | 5 µM |
| Chemical compound, drug | N-Acetyl-L-cysteine | Sigma-Aldrich | Cat# A7250 | 5 mM |
| Chemical compound, drug | Quercetin | Sigma-Aldrich | Cat# Q4951 | Cell (50 µM), Animal (50 µg/g body weight) |
| Chemical compound, drug | Dasatinib | Sigma-Aldrich | Cat# CDS023389 | Cell (500 nM), Animal (5 µg/g body weight) |
| Chemical compound, drug | Navitoclax (ABT-263) | Selleck | Cat# S1001 | 10 ug/g body weight |
| Software, algorithm | Bowtie 2 | PMID:22388286 | RRID:SCR_016368 | Version 2.2.6 |
| Software, algorithm | Systems Transcriptional Activity Reconstruction (STAR) | PMID:23104886 | RRID:SCR_005622 | Version 2.5.3 |
| Software, algorithm | Cuffdiff | PMID:23222703 | RRID:SCR_001647 | Version 2.2.1 |
| Software, algorithm | Ingenuity Pathway Analysis | Qiagen | RRID:SCR_008653 | |
| Software, algorithm | PANTHER | http://www.pantherdb.org/ | RRID:SCR_004869 | Version 14.1 |
| Software, algorithm | Gene Set Enrichment Analysis (GSEA) | http://www.broadinstitute.org/gsea/ | RRID:SCR_003199 | Version 4.3.3 |
| Software, algorithm | ConsensusPathDB | http://cpdb.molgen.mpg.de | RRID:SCR_002231 | Version 14.10.2021 |
| Software, algorithm | MaxQuant | http://www.biochem.mpg.de/5111795/maxquant | RRID:SCR_014485 | Version 1.6.2.10 |
| Software, algorithm | Fiji | PMID:22743772 | RRID:SCR_002285 | Version 2.16.0 |
| Software, algorithm | Flowjo | BD Biosciences | RRID:SCR_008520 | Version 10.10.0 |
| Software, algorithm | GraphPad Prism | GraphPad Prism (https://graphpad.com) | RRID:SCR_015807 | Version 10.1.1 |

## Mice

In this study, two *Chmp5*-flox mouse strains were used, and similar phenotypes were observed. The Chmp5<sup>tm2.1Gho</sup> strain was previously described (*Greenblatt et al., 2015*). The alternative *Chmp5*-flox strain was established by CRISPR/Cas9-mediated genome engineering. Similarly to the Chmp5<sup>tm2.1Gho</sup> strain, exons 4 and 5 were selected as the conditional knockout region in the newly created strain, and deletion of this region results in a frameshift of the *Chmp5* gene. To engineer the target vector, homologous arms, and the conditional knockout region with 5' and 3' loxP sites were generated by PCR using the BAC clone. Cas9, gRNA, and targeting vectors were co-injected into fertilized eggs for gene-edited mouse production, and pups were genotyped by PCR followed by sequencing analysis. Mice bearing the targeted allele were backcrossed with C57BL/6 J mice up to the F8 generation.

Ctsk<sup>tm1(cre)Ska</sup> (Ctsk<sup>Cre</sup>), B6N.FVB-Tg(Dmp1-cre)1Jqfe/BwdJ (Dmp1-Cre), B6.Cg-Tg(Prrx1-cre)1Cjt/J (Prrx1<sup>Cre</sup>), B6;SJL-Tg(Col2a1-cre)1Bhr/J (Col2a1-Cre), and B6.Cg-Tg(Sp7-tTA,tetO-EGFP/cre)1Amc/J (Osx1-GFP::Cre) mice have previously been reported (*Greenblatt et al., 2015*; *Logan et al., 2002*; *Lu et al., 2007*; *Rodda and McMahon, 2006*; *Nakamura et al., 2007*). While the Ctsk-Cre strain is a knock-in mouse line, other strains (Dmp1-Cre, Prrx1-Cre, Col2a1-Cre, and Osx-Cre) are transgenic lines. The Gt(ROSA)26Sor<sup>tm4(ACTB-tdTomato,-EGFP)Luo</sup> (Rosa26<sup>mTmG</sup>) mice were purchased from Jackson Laboratories. To generate experimental animals, *Chmp5<sup>fl/fl</sup>* mice were crossed with *Ctsk<sup>Cre</sup>;Chmp5<sup>fl/+</sup>* and *Dmp1<sup>Cre</sup>;Chmp5<sup>fl/+</sup>* mice, respectively. To generate reporter mouse lines, *Ctsk<sup>Cre</sup>;Chmp5<sup>fl/+</sup>* mice were mated with *Chmp5<sup>fl/fl</sup>;Rosa26<sup>mTmG</sup>* mice, and *Ctsk<sup>Cre</sup>* mice were crossed with *Rosa26<sup>mTmG</sup>* animals.

All mice were kept in a C57BL/6 or a C57BL/6/129S6 mixed background and housed in the standard animal facility on a 12 hr light/dark cycle with ad libitum access to water and standard chow. In most situations, the littermates were compared otherwise as indicated. All animals were used in accordance with the Guide for Care and Use of Laboratory Animals (NIH, United States) and were handled

according to protocols approved by the IACUC at the University of Massachusetts Medical School (Worcester, United States) and Capital Medical University (Beijing, China). The biological and technical replicates (*n* numbers) for each animal experiment are reported in the figure legends.

In this study, both sexes of mice were used, and unless otherwise stated, no gender differences were observed in the reported phenotypes. No animals were excluded from all analyses unless they had unexpected deaths or illnesses unrelated to this study. Due to the overt musculoskeletal phenotypes of *Ctsk^Cre;Chmp5^fl/fl* and *Dmp1^Cre;Chmp5^fl/fl* mice, the experimenters could not be blinded to animal genotypes. All data were analyzed by individuals with no knowledge of the experimental groups.

## Micro-CT and radiography

Micro-CT analyses were performed using the Scanco µCT-35 or µCT-40 scanner (Scanco Medical). Images were processed using the Inveon Research Workplace (Siemens Medical Solutions). X-ray images were acquired using the X-Ray MX-20 Specimen (Faxitron) following the default programme.

## Histology, immunohistochemistry, and immunofluorescence

All samples were fixed in 10% neutral formalin or 4% paraformaldehyde overnight for premature samples or 2 d for mature skeletal samples. Samples were decalcified with 14% EDTA and serial sections were cut for all samples. At least five slides were selected at a certain interval (80–100 µm) from each sample for histological evaluation. Fate-mapping images were acquired using a Leica TCS SP5 II laser scanning confocal microscope.

Immunofluorescence for γH2AX was performed using a Phospho-Histone H2A.X (Ser139) antibody (clone 20E3, CST). Immunohistochemistry for Ki-67 was performed using the VECTASTAIN Elite ABC HRP Kit (Peroxidase, Rabbit IgG; Vector Laboratories). Cells for immunofluorescence were cultured in 35 mm MatTek glass bottom dishes (No. 1.5 coverslip, MatTek Corporation). Briefly, tissues or cells were fixed in 4% PFA for 15 min, blocked in 1% BSA for 30 min, and incubated with primary antibodies (mouse anti-EEA1 monoclonal antibody, rabbit anti-RAB7 monoclonal antibody, rabbit anti-RAB11 monoclonal antibody, rat anti-CD107a monoclonal antibody, rat anti-TGN38 monoclonal antibody) for 2 hr and then fluorescence-conjugated secondary antibodies for 1 hr. Images were acquired using the Leica TCS SP5 II laser scanning confocal microscope. The quantification of intracellular vesicles was performed using FIJI software following the particle analysis procedure (https://imagej.net/imagej-wiki-static/Particle_Analysis, NIH).

## Bone mechanical test

The femurs of *Ctsk^Cre;Chmp5^fl/fl* and *Chmp5^fl/fl* mice at 6–7 wk of age were subjected to mechanical testing using the DMA Q800 mechanical test system with the 3-point bending mode following the manufacturer's procedure (TA Instruments, USA). The femurs were placed in a direction of the anterior surface downward and the posterior aspect of the femoral condyles facing upward. The span length (L) was 10 mm and the load was applied in the middle of the bone. Five mice per group were used and bone stiffness and ultimate stress (fracture stress) from male mice were reported. Similar changes were observed in both sexes.

## Tests of skeletal muscle functions

The four-limb hanging test was performed using the top grid of a mouse cage (*Bonetto et al., 2015*). The grid was set at a height of about 35 cm, and the measurement was repeated for 3–5 times for each mouse at a rest interval of about 3–5 min. The maximum hang time was recorded for each animal. Both male and female animals were included in this experiment, and there was no significant sex difference regarding the four-limb hanging time.

The forelimb grip strength test was carried out using an electronic scale as described with slight modification (*Bonetto et al., 2015*; *Takeshita et al., 2017*). A mouse was allowed to grasp the gauze attached to the scale. The scale was reset to 0 g after stabilization and a researcher slowly pulled the mouse's tail backward. Five consecutive measurements were performed for each mouse and the peak pull force in grams was recorded. A significant sex difference was observed in the forelimb grip strength. Both male and female mice were included in this study, and the same trend of changes was found in both genders.

## Skeletal progenitor cell culture and sorting

The periskeletal progenitors were isolated from the hindlimbs of *Ctsk^Cre^;Rosa26^mTmG/+^*, *Ctsk^Cre^;Chmp-5^fl/+^;Rosa26^mTmG/+^*, and *Ctsk^Cre^;Chmp5^fl/fl^;Rosa26^mTmG/+^* mice (n=10 animals per genotype) at 2 wk of age according to the previously established method (*Ge et al., 2016*). Briefly, the periskeletal tissues were dissected, minced into small pieces, and digested with 1 mg/ml collagenase type I (Worthington), 1 mg/ml collagenase type II (Worthington), and 1.5 mg/ml dispase (Roche) for 30 min at 37 °C. Cells were cultured in skeletal stem cell medium (Joint Therapeutics, Beijing, China) for 1 wk, stained with anti-CD31 and anti-CD45 antibodies, and sorted using a FACSAria II cell sorter (BD Biosciences) for further analyses.

In all experiments, *Ctsk^Cre^;Chmp5^fl/fl^* and control cells from littermate mice were cultured, sorted, and re-seeded in parallel for subsequent analyses. The coculture experiment was carried out by directly mixing 90% wild-type skeletal progenitors with 10% *Ctsk^Cre^;Chmp5^fl/fl^* or control periskeletal progenitors to simulate the in vivo context of cell-cell contact in periskeletal overgrowth. The cocultures were induced in osteogenic differentiation medium for 4 wk. The biological and technical replicates (*n*-numbers) for each cellular experiment are reported in the figure legends.

## CRISPR/CAS9 lentiviral infection

Pairs of CRISPR guide RNA oligos (mouse *Chmp5* single guide RNAs [sgRNAs] targeting GGCTCCGC CACCTAGCTTGA and GTTTCGCTTTTCCGAAGAAT on exon 1, respectively) were annealed and cloned into the BsmBI sites of the lentiCRISPR V2-puro vector (plasmid 52961, Addgene). CRISPR lentiviral plasmids and lentiviral packaging plasmids (pMDLg/pRRE, pRSV-Rev, and pMD2.G; Addgene) were transfected into HEK 293T cells. The supernatants were harvested and filtered through a 0.45 µm filter 2.5 d after transfection. MC3T3-E1 and ATDC5 cells were infected with CRISPR lentivirus and selected with puromycin (4.5 µg/ml, Clontech) for 7 d. The depletion of *Chmp5* was confirmed by Western blotting.

## In vitro osteogenesis

For osteogenic differentiation, $1.0×10^5$ cells were seeded in each well of 24-well plates and cultured overnight. Cells were changed into osteogenic media (Joint Therapeutics) the next day and induced for the indicated times. Osteogenesis was determined by alizarin red and von Kossa staining and by examining the activity of alkaline phosphatase.

## Cell cycle, proliferation, and apoptosis analyses

The cell number was counted with a Countess II FL Automated Cell Counter (Thermo Fisher Scientific, USA) at the indicated time intervals. The AlamarBlue cell viability assay was performed as previously described (*Ge et al., 2016*). The cell cycle was determined using the APC-BrdU Flow Kit (BD Biosciences). In vitro apoptosis analysis was carried out with Annexin V-PI staining (ThermoFisher Scientific), and in situ cell apoptosis was determined with TUNEL staining (Abcam).

## RNA-seq, data analysis, and functional annotation

The sorted *Ctsk^Cre^;Chmp5^fl/fl^* and *Ctsk^Cre^;Chmp5^fl/+^* periskeletal progenitors were subjected to RNA-seq. For library preparation, 100 ng of total RNA from each sample was subjected to rRNA depletion with the NEBNext rRNA Depletion Kit (New England Biolabs) according to the manufacturer's manual. Subsequently, the rRNA-depleted RNA was used to build the RNA-seq library with the NEBNext Ultra II Directional RNA Library Prep Kit (New England Biolabs) according to the manufacturer's manual. The index of each RNA-seq library was introduced using NEBNext Multiplex Oligos for Illumina (New England Biolabs). Each RNA-seq library was analyzed by the fragment analyzer (Advanced Analytical Technologies) and quantified by the KAPA Library Quantification Kit (KAPA Biosystems) according to the manufacturer's manual. An equal amount of each RNA-seq library was mixed for sequencing in one lane of Illumina HiSeq3000 in paired-end 150 base mode (Illumina).

The raw Illumina pair-end reads were first aligned with ribosomal RNA (GenBank ID BK000964.1) with Bowtie2 v2.2.6 (*Langmead and Salzberg, 2012*). Reads that failed to map to ribosomal RNA were aligned with the mouse reference genome mm10 with STAR v2.5.3 (*Dobin et al., 2013*). Differential analysis was performed with Cuffdiff v2.2.1 (*Trapnell et al., 2013*). The raw data were analyzed by two independent statisticians and similar results were obtained.

Pathway analyses were performed for differentially expressed genes using the Reactome online analysis tool (*Fabregat et al., 2017*) and the Ingenuity Pathway Analysis software (Qiagen). Gene ontology term over-representation analysis was run using PANTHER (version 14.1) with all genes expressed in a tissue or cell type as background and the differentially expressed gene list as input. The P-values of Fisher's exact tests were corrected by controlling the false discovery rate (FDR). Gene set enrichment analyses were performed using GSEA or ConsensusPathDB. For GSEA-based analysis, gene sets of interest were downloaded from the MsigDB, while all built-in gene sets were used for ConsensusPathDB-based analysis.

## Nano LC-MS/MS for secretomic and proteomic analyses

Label-free relative protein quantification analyses were performed to identify differentially expressed proteins in the supernatant or cell lysis of $Ctsk^{Cre};Chmp5^{fl/fl}$ versus control periskeletal progenitors (n=3 animals per group). For cell supernatants, the same amounts of samples went through 3 kD ultrafiltration tubes and then were processed for reduction and alkylation. Subsequently, the samples were digested in trypsin overnight, and the peptides were desalted prior to mass spectrometry analysis. For cell lysis, samples were precipitated using acetone after reduction and alkylation. The samples were then subjected to digestion, desalination, and mass spectrometry analysis.

Quantitative data were collected on an Orbitrap Eclipse Tribrid mass spectrometer (Thermo Fisher Scientific) coupled to a nanoLC system (EASY-nLC 1200, Thermo Fisher Scientific). All peptide mixtures were analyzed using the same chromatographic conditions: 5 µl of each sample were fractionated in a 150 µm×15 cm in-house made column packed with Acclaim PepMap RPLC C18 (1.9 µm, 100 Å, Dr. Maisch GmbH, Germany) working at a flow rate of 600 nl/min, using a linear gradient of eluent B (20–0.1% formic acid in water +80% acetonitrile) in A (0.1% formic acid in MilliQ water) from 4 to 10% for 5 min, 10 to 22% for 80 min, 22 to 40% for 25 min, 40 to 95% for 5 min, and 95 to 95% for 5 min. MS/MS analyses were performed using Data-Dependent Acquisition (DDA) mode: one MS scan (mass range from 350 to 1500 m/z) was followed by MS/MS scans up to the top 20 most intense peptide ions from the preview scan in the Orbitrap, applying a dynamic exclusion window of 120 s.

The raw MS files were analyzed and searched against the Mus_musculus_reviewed protein database using MaxQuant (1.6.2.10). The parameters were set as follows: the protein modifications were carbamidomethylation (C) (fixed), oxidation (M) (variable), acetyl (Protein N-term) (variable); the enzyme specificity was set to trypsin; the maximum missed cleavages were set at 2; the precursor ion mass tolerance was set at 20 ppm; and the MS/MS tolerance was 20 ppm. Only highly confident identified peptides were chosen for downstream protein identification analysis.

## Western blotting

Western blot was performed using 4–20% Mini-PROTEAN TGX precast protein gels (Bio-Rad). The following antibodies: anti-CHMP5 polyclonal antibody (*Shim et al., 2006*), anti-COL1A1 monoclonal antibody (clone E8F4L, Cell Signaling Technology, CST), anti-TAGLN polyclonal antibody (cat#: AB14106, Abcam), anti-p16 INK4A monoclonal antibody (clone E5F3Y, CST), anti-p21 Waf1/Cip1 monoclonal antibody (clone E2R7A, CST), anti-VPS4A monoclonal antibody (clone A-11, Santa Cruz Technology), total OXPHOS antibody cocktail (Abcam), anti-GAPDH monoclonal antibody (clone 14C10, CST), and anti-β-Actin monoclonal antibody (clone D6A8, CST), were used.

## Quantitative PCR

Total RNA was isolated from cells using Trizol reagent (Qiagen). RNA samples were treated with the TURBO DNA-free Kit (Thermo Fisher) and equal amounts (1–2 µg) were used for reverse transcriptase reaction using an iScript Reverse Transcription Supermix (Bio-Rad). Quantitative PCR was run using the SYBR Green Master Mix on a CFX Connect Real-Time PCR Detection System (Bio-Rad). Gene expression levels were analyzed relative to the housekeeping gene *Gapdh* and presented by $2^{-\Delta Ct}$ (*Schmittgen and Livak, 2008*). The primer sequences were used for qPCR are as follows: *Chmp5*, forward, 5'-ATGAGAGAGGGTCCTGCTAAG-3', reverse, 5'-CCGTGGTCTTGGTGTCCTTTA-3'; *H2afx*, forward, 5'-GTGGTCTCTCAGCGTTGTTC-3', reverse, 5'-CGGCC TACAGGGAACTGAA-3'; *Tp53*, forward, 5'-GTCACAGCACATGACGGAGG-3', reverse, 5'-TCTTCCAGATGCTCGGGATAC-3'; *Gapdh*, forward 5'-TGCCAGCCTCGTCCCG TAGAC-3', reverse 5'-CCTCACCCCATTTGATGTTAG-3'; *VPS4A*, forward 5'- AGAACCAGAGTGAGGGCAAGGG-3', reverse 5'- GCACCCATCAGCTGTTCTTGCA-3';

*CHMP5*, forward 5'-CCAGCCTGACTGACTGCATTGG-3', reverse 5'- TCGCAAGGCTTTCTGCTTGACC-3'; *GAPDH*, forward 5'- GGAGTCCACTGGCGTCTTCAC-3', reverse 5'-GAGGCATTGCTGATGATCTTGAGG-3'.

## Extracellular vesicle tracking

The concentration and size distribution of extracellular vesicles were analyzed using the NanoSight NS300 following the manufacturer's protocol (Malvern Instruments). Briefly, cells were seeded in 24-well plates at the density of $1 \times 10^5$ cells per well and cultured overnight. The next morning, cells were changed to serum-free medium and incubated for 9 hr. The medium was collected and subjected to sequential centrifugation at 300 g for 10 min, 2000 g for 20 min, and 10,000 g for 30 min. The resulting supernatant was manually injected into the instrument and run in the 'standard measurement' module with five captures per sample and the data were processed with a detection threshold of 2.

## Live-cell imaging for cell endocytosis

The wild-type and *Ctsk^Cre^;Chmp5^fl/fl^* periskeletal progenitors were incubated on ice for 10 min before adding fresh medium with 2 ug/ml pHrodo red EGF conjugate (Thermo Fisher). Cells were incubated at 37 °C for 30 min and subsequently washed and changed to fresh medium. Live-cell images were obtained at 1, 12, and 24 hr using the Leica TCS SP5 II laser scanning confocal microscope.

## Transmission electron microscopy

Wild-type and *Ctsk^Cre^;Chmp5^fl/fl^* periskeletal progenitors were fixed in culture plates overnight at 4 °C using 2.5% glutaraldehyde in 0.1 M Na cacodylate-HCl buffer (pH 7.2). After washing in 0.5 M Na cacodylate-HCl buffer (pH 7.0), the cells were post-fixed for 1 hr in 1% osmium tetroxide (w/v) at room temperature. After post-fixation, the culture plate with adherent cells was en bloc stained (20 min) with 1% aqueous uranyl-acetate (w/v). Culture plates were washed again in the same buffer and dehydrated through a series of graded ethanol to 100% and transferred through two changes of 50/50 (v/v) SPIpon resin (Structure Probe, Inc)/100% ethanol and left overnight to infiltrate. The following morning, the cell culture plates were transferred through three changes of fresh SPIpon resin to finish the infiltration and embedding and finally, the plates were filled with freshly prepared SPIpon resin and polymerized for 48 hr at 70 °C.

Once fully polymerized, the plate was cut apart, and each well was plunged into liquid nitrogen to separate the SPIpon epoxy block with the embedded cells from the culture dish. The round epoxy disks with the embedded cells were then examined under an upright light microscope, and areas of cells were cut from the disks and glued onto blank microtome studs and trimmed for ultramicrotomy. Ultrathin sections (70 nm) were cut on a Reichert-Jung ultramicrotome using a diamond knife. The sections were collected and mounted on copper support grids and contrasted with lead citrate and uranyl acetate, and examined on a FEI Tecnai G2 Spirit transmission electron microscope at 100 kV accelerating voltage and images were recorded at various magnifications using a Gatan 2 K digital camera system.

## Mitochondrial respiration

Mitochondrial respiration was examined in *Ctsk^Cre^;Chmp5^fl/fl^* and wild-type periskeletal progenitors using a Seahorse XF^e^96 Extracellular Flux Analyzer (Agilent) as previously described (*Fiorese et al., 2016*). Briefly, 10,000 cells per well were seeded in XF 96-well culture plates and cultured overnight. The next day, cells were changed to assay medium and incubated at 37 °C w/o CO2 for 1 hr. The assay was carried out using the Seahorse XF Cell Mito Stress Test kit with sequential injections of Oligomycin (1 μM), FCCP (1 μM), and Rotenone plus Antimycin A (0.5 μM) following the manufacturer's protocol.

## Senolytic treatments

For in vitro cell treatment, *Ctsk^Cre^;Chmp5^fl/fl^* and control cells were seeded in 96-well plates at a density of 6000 cells/ well and cultured overnight. Quercetin and dasatinib were added to the culture medium at a final concentration of 50 μM and 500 nM, respectively. Apoptotic cells are labeled using Incucyte Annexin V Red Dye and are quantified in real time using the Incucyte Live-Cell Analysis System. Six

replicates were used for each group and the experiment was repeated twice using cells from two different animals per group.

For in vivo treatment, the senolytic drugs quercetin and dasatinib were administered intraperitoneally weekly at 50 μg/g and 5 μg/g body weight, respectively. *Senolytic* treatment in *Ctsk^Cre;Chmp5^fl/fl* and littermate control mice started from the first week after birth, and mice were collected at 7–8 wk of age for skeletal analyses. For the *Dmp1^Cre;Chmp5^fl/fl* strain, treatment began the second week after birth, and the animals were monitored for 16 wk. ABT-263 was administered intraperitoneally to *Ctsk^Cre;Chmp5^fl/fl* and littermate control mice weekly at 10 μg/g body weight, starting from the first week after birth, and mice were collected at 8 wk of age for micro-CT analyses. In these in vivo studies, mice were randomly assigned to receive the corresponding senolytic treatment and evaluated blindly to experimental conditions. There was no attrition of animals during these experimental procedures.

## Statistics

All data are represented as mean ±s.d. Normality and equal variance of the data sets were tested using the Shapiro-Wilk test and the F test, respectively. The data were determined to be normally distributed and have equal variance unless otherwise specified. For experiments with three or more groups, statistical analysis was performed using one-way or two-way ANOVA followed by multiple comparisons. For comparisons between two groups, the two-tailed unpaired Student's *t*-test was applied. All analyses were performed with Prism 10.1.1 (GraphPad).

## Acknowledgements

We thank Dr. Sankar Ghosh for providing Chmp5-floxed mice; Drs. Gregory Hendricks and Lara Strittmatter at the University of Massachusetts Chan Medical School for help in interpreting TEM data; Dr. Tomer Shpilka for assistance in running the Seahorse analyzer; Drs. Guangchuang Yu and Jiyeon Park for independently analyzing the RNA-seq data. Dr. Chunjing Bian for generating the CHMP5 overexpression vector. X.G. is supported by the National Natural Science Foundation of China (Science Fund Program for Distinguished Young Scholars (Overseas)) and the Beijing Natural Science Foundation (project 5222008). J.H.S. holds support from the NIH (R21AR077557, R01AR078230) and AAVAA Therapeutics.

## Additional information

### Competing interests

Fan Zhang: Employee of Joint Therapeutics Co. Ltd. Jae-Hyuck Shim: is a scientific co-founder of AAVAA Therapeutics and holds equity in this company but does not have conflicts of interest with this study. The other authors declare that no competing interests exist.

### Funding

| Funder | Grant reference number | Author |
|---|---|---|
| National Natural Science Foundation of China | Science Fund Program for Distinguished Young Scholars (Overseas) | Xianpeng Ge |
| Beijing Natural Science Foundation | 5222008 | Xianpeng Ge |
| National Institute of Arthritis and Musculoskeletal and Skin Diseases | R21AR077557 | Jae-Hyuck Shim |
| National Institute of Arthritis and Musculoskeletal and Skin Diseases | R01AR078230 | Jae-Hyuck Shim |

| Funder | Grant reference number | Author |
|--------|------------------------|--------|

The funders had no role in study design, data collection and interpretation, or the decision to submit the work for publication.

## Author contributions

Fan Zhang, Validation, Investigation, Writing – review and editing; Yuan Wang, Luyang Zhang, Chunjie Wang, Investigation; Deping Chen, Ren Xu, Resources; Haibo Liu, Formal analysis, Visualization, Methodology; Cole M Haynes, Resources, Methodology; Jae-Hyuck Shim, Resources, Funding acquisition, Writing – review and editing; Xianpeng Ge, Conceptualization, Resources, Data curation, Formal analysis, Supervision, Funding acquisition, Validation, Investigation, Visualization, Methodology, Writing – original draft, Project administration, Writing – review and editing

## Author ORCIDs

Fan Zhang ⓘ https://orcid.org/0009-0006-6245-0199
Haibo Liu ⓘ https://orcid.org/0000-0002-4213-2883
Ren Xu ⓘ https://orcid.org/0000-0001-6578-4553
Cole M Haynes ⓘ https://orcid.org/0000-0003-2110-5648
Jae-Hyuck Shim ⓘ https://orcid.org/0000-0002-4947-3293
Xianpeng Ge ⓘ https://orcid.org/0000-0002-1291-2096

## Ethics

All animals were used in accordance with the Guide for Care and Use of Laboratory Animals (NIH, United States) and were handled according to protocols approved by IACUC at the University of Massachusetts Medical School (Worcester, United States) and Capital Medical University (Beijing, China; protocol #: AEEI-2022-036).

Reviewer #1 (Public review): https://doi.org/10.7554/eLife.101984.3.sa1
Reviewer #3 (Public review): https://doi.org/10.7554/eLife.101984.3.sa2
Author response https://doi.org/10.7554/eLife.101984.3.sa3

---

# Additional files

## Supplementary files

MDAR checklist

## Data availability

All data generated or analyzed during this study are included in the manuscript and supplemental files. The raw datasets for the transcriptome (https://doi.org/10.5061/dryad.z8w9ghxrk), secretome (https://doi.org/10.5061/dryad.0rxwdbscp), and proteome (https://doi.org/10.5061/dryad.cfxpnvxk2) are available in the Dryad Digital Repository. All experimental materials in this study are available on reasonable request to the corresponding author.

The following datasets were generated:

| Author(s) | Year | Dataset title | Dataset URL | Database and Identifier |
|-----------|------|---------------|-------------|-------------------------|
| Ge X | 2025 | Transcriptome dataset for: The ESCRT protein CHMP5 restricts bone formation by controlling endolysosome-mitochondrion-mediated cell senescence | https://doi.org/10.5061/dryad.z8w9ghxrk | Dryad Digital Repository, 10.5061/dryad.z8w9ghxrk |

*Continued on next page*

*Continued*

| Author(s) | Year | Dataset title | Dataset URL | Database and Identifier |
|-----------|------|---------------|-------------|-------------------------|
| Ge X | 2025 | Secretome dataset for: The ESCRT protein CHMP5 restricts bone formation by controlling endolysosome-mitochondrion-mediated cell senescence | https://doi.org/10.5061/dryad.0rxwdbscp | Dryad Digital Repository, 10.5061/dryad.0rxwdbscp |
| Ge X | 2025 | Proteomic dataset for: The ESCRT protein CHMP5 restricts bone formation by controlling endolysosome-mitochondrion-mediated cell senescence | https://doi.org/10.5061/dryad.cfxpnvxk2 | Dryad Digital Repository, 10.5061/dryad.cfxpnvxk2 |

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
