## [Editor Report · eLife Assessment]

This **important** work advances our understanding of CHMP5's role in regulating osteogenesis through its impact on cellular senescence. The evidence supporting the conclusion is **convincing** and the revised manuscript is largely improved. This paper holds potential interest for skeletal biologists who study the pathogenesis of age-associated skeletal disorders.

---

## [Referee Report · Reviewer #1 (Public review)]

Summary:

The manuscript presents a significant and rigorous investigation into the role of CHMP5 in regulating bone formation and cellular senescence. The study provides compelling evidence that CHMP5 is essential for maintaining endolysosomal function and controlling mitochondrial ROS levels, thereby preventing the senescence of skeletal progenitor cells.

Strengths:

The authors demonstrate that the deletion of Chmp5 results in endolysosomal dysfunction, elevated mitochondrial ROS, and ultimately enhanced bone formation through both autonomous and paracrine mechanisms. The innovative use of senolytic drugs to ameliorate musculoskeletal abnormalities in Chmp5-deficient mice is a novel and critical finding, suggesting potential therapeutic strategies for musculoskeletal disorders linked to endolysosomal dysfunction.

Comments on the latest version:

My concerns were addressed.

---

## [Referee Report · Reviewer #3 (Public review)]

Summary:

In this study, Zhang et al. reported that CHMP5 restricts bone formation by controlling endolysosome-mitochondrion-mediated cell senescence. Zhang et al., report a novel role of CHMP5 on osteogenesis through affecting cell senescence. Overall, it is an interesting study and provides new insights in the field of cells senescence and bone.

Strengths:

Analyzed the bone phenotype OF CHMP5-periskeletal progenitor-CKO mouse model and found the novel role of senescent cells on osteogenesis and migration.

Weaknesses:

(1) The role and mechanism of CHMP5 gene deletion in enhancing osteogenesis via cellular senescence remain insufficiently elucidated.

(2) The use of the ADTC5 cell line as a skeletal precursor/progenitor model is suboptimal.

Overall, the results support their conclusions.

The impact of this work on the field is its proposal that cellular senescence may exert either inhibitory or promotive effects on osteogenic capacity, depending on cell type and context.

The revised manuscript has addressed most of the concerns raised during the initial review.

---

## [Author Response]

The following is the authors’ response to the original reviews

**Public Reviews:**

**Reviewer #1 (Public review):**
Summary:The manuscript presents a significant and rigorous investigation into the role of CHMP5 in regulating bone formation and cellular senescence. The study provides compelling evidence that CHMP5 is essential for maintaining endolysosomal function and controlling mitochondrial ROS levels, thereby preventing the senescence of skeletal progenitor cells.Strengths:The authors demonstrate that the deletion of Chmp5 results in endolysosomal dysfunction, elevated mitochondrial ROS, and ultimately enhanced bone formation through both autonomous and paracrine mechanisms. The innovative use of senolytic drugs to ameliorate musculoskeletal abnormalities in Chmp5-deficient mice is a novel and critical finding, suggesting potential therapeutic strategies for musculoskeletal disorders linked to endolysosomal dysfunction.Weaknesses:The manuscript requires a deeper discussion or exploration of CHMP5's roles and a more refined analysis of senolytic drug specificity and effects. This would greatly enhance the comprehensiveness and clarity of the manuscript.

We thank the reviewer for these insightful comments. In the revised manuscript, we have expanded the discussion of the distinct roles of CHMP5 in different cell types. Specifically, we add the following sentences (Lines 433-439 in the combined manuscript):

“Also, a previous study by Adoro et al. did not detect endolysosomal abnormalities in Chmp5 deficient developmental T cells [1]. Since both osteoclasts and T cells are of hematopoietic origin, and meanwhile osteogenic cells and MEFs, which show endolysosomal abnormalities after CHMP5 deficiency, are of mesenchymal origin, it turns out that the function of CHMP5 in regulating endolysosomal pathway could be cell lineage-specific, which remains clarified in future studies.”

In addition, we tested another senolytic drug Navitoclax (ABT-263), which is a BCL-2 family inhibitor and induces apoptosis of senescent cells, in *Chmp5Ctsk* mice. Micro-CT analysis showed that ABT-263 could also improve periskeletal bone overgrowth in *Chmp5Ctsk* mice (Fig. 5F). Furthermore, we have also discussed the potential off-target effects of senolytic drugs in *Chmp5Ctsk* mice in the revised manuscript. Specifically, we added the following paragraph (Lines 441-451):

“Furthermore, it is unclear whether the effect of senolytic drugs in *Chmp5Ctsk* mice involves targeting osteoclasts other than osteogenic cells, as osteoclast senescence has not yet been evaluated. However, the efficacy of Q + D in targeting osteogenic cells, which is the focus of the current study, was confirmed in *Chmp5Dmp1* mice (Fig. 5C-E). Additionally, Q + D caused a higher cell apoptotic ratio in *Chmp5Ctsk* compared to *wild-type* periskeletal progenitors in ex vivo culture (Fig. 5A), demonstrating the effectiveness of Q + D in targeting osteogenic cells in the *Chmp5Ctsk* model. Furthermore, an alternative senolytic drug ABT-263 could also ameliorate periskeletal bone overgrowth in *Chmp5Ctsk* mice (Fig. 5F). Together, these results confirm that osteogenic cell senescence is responsible for the bone overgrowth in *Chmp5Ctsk* and *Chmp5Dmp1* mice, and senolytic treatments are effective in alleviating these skeletal disorders.”

**Reviewer #2 (Public review):**
Summary:The authors try to show the importance of CHMP5 for skeletal development.Strengths:The findings of this manuscript are interesting. The mouse phenotypes are well done and are of interest to a broader (bone) field.Weaknesses:The mechanistic insights are mediocre, and the cellular senescence aspect poor.In total, it has not been shown that there are actual senescent cells that are reduced after D+Qtreatment. These statements need to be scaled back substantially.

We thank the reviewer for these suggestive comments. We have added additional results to strengthen the senescent phenotypes of *Chmp5*-deficient skeletal progenitor cells, including significant enrichment of the SAUL_SEN_MAYO geneset (positively correlated with cell senescence) and the KAMMINGA_SENESCENCE geneset (negatively correlated with cell senescence) at the transcriptional level by GSEA analysis of RNA-seq data (Fig. S3C), and the increase of γH2Ax^+^;GFP^+^ cells at periskeletal overgrowth in *Chmp5Ctsk;Rosa2626mTmG/+* mice vs. the periosteum of *Chmp5Ctsk/+;Rosa2626mTmG/+* control mice (Fig. 3E). These results further advocate for the senescent phenotypes of *Chmp5*-deficient skeletal progenitors.

Furthermore, the combination of Q + D caused a higher cell apoptotic ratio in *Chmp5Ctsk* vs. *wildtype* periskeletal progenitors in ex vivo culture (Fig. 5A), suggesting their effectiveness in targeting periskeletal progenitor cell senescence in *Chmp5Ctsk* mice. Furthermore, we tested an alternative senolytic drug ABT-263, which is an inhibitor of the BCL-2 family and induces apoptosis of senescent cells, in *Chmp5Ctsk* mice, and ABT-263 could also alleviate periskeletal bone overgrowth in *Chmp5Ctsk* mice (Fig. 5F). Together, these results demonstrate that osteogenic cell senescence is responsible for abnormal bone overgrowth in *Chmp5*-deficient mice and that senolytic drugs are effective in improving these skeletal disorders.

**Reviewer #3 (Public review):**
Summary:In this study, Zhang et al. reported that CHMP5 restricts bone formation by controlling endolysosomemitochondrion-mediated cell senescence. The effects of CHMP5 on osteoclastic bone resorption and bone turnover have been reported previously (PMID: 26195726), in which study the aberrant bone phenotype was observed in the CHMP5-ctsk-CKO mouse model, using the same mouse model, Zhang et al., report a novel role of CHMP5 on osteogenesis through affecting cell senescence. Overall, it is an interesting study and provides new insights in the field of cell senescence and bone.Strengths:Analyzed the bone phenotype OF CHMP5-periskeletal progenitor-CKO mouse model and found the novel role of senescent cells on osteogenesis and migration.Weaknesses:(1) There are a lot of papers that have reported that senescence impairs osteogenesis of skeletal stem cells. In this study, the author claimed that Chmp5 deficiency induces skeletal progenitor cell senescence and enhanced osteogenesis. Can the authors explain the controversial results?

Different skeletal stem cell populations in time and space have been identified and reported [2-6]. The present study shows that *Chmp5* deficiency in periskeletal (*Ctsk*-Cre) and endosteal (*Dmp1*-Cre) osteogenic cells causes cell senescence and aberrant bone formation. Although cell senescence during aging can impair the osteogenesis of marrow stromal cells (MSCs), which contributes to diseases with low bone mass such as osteoporosis, aging can also increase heterotopic ossification or mineralization in musculoskeletal soft tissues such as ligaments and tendons [7]. Notably, the abnormal periskeletal bone overgrowth in *Chmp5Ctsk* mice was mainly mapped to insertion sites of tendons and ligaments on the bone (Fig. 1A and E), consistent with changes during aging. More broadly, aging can also cause abnormal ossification or mineralization in other body tissues, such as the heart valve [8, 9]. These different results reflect an aberrant state of ossification or mineralization in musculoskeletal tissues and throughout the body during aging. Based on the reviewer’s comment, we have discussed these results in the revised manuscript. Specifically, we add the following paragraph (Lines 453-462 in the combined manuscript):

“Notably, aging is associated with decreased osteogenic capacity in marrow stromal cells, which is related to conditions with low bone mass, such as osteoporosis. Rather, aging is also accompanied by increased ossification or mineralization in musculoskeletal soft tissues, such as tendons and ligaments [7]. In particular, the abnormal periskeletal overgrowth in *Chmp5Ctsk* mice was predominantly mapped to insertion sites of tendons and ligaments on the bone (Fig. 1A and E), which is consistent with changes during aging and suggests that mechanical stress at these sites could contribute to the aberrant bone growth. These results suggest that skeletal stem/progenitor cells at different sites of musculoskeletal tissues could demonstrate different, even opposite outcomes in osteogenesis, due to cell senescence.”

(2) Co-culture of Chmp5-KO periskeletal progenitors with WT ones should be conducted to detect the migration and osteogenesis of WT cells in response to Chmp5-KO-induced senescent cells. In addition, the co-culture of WT periskeletal progenitors with senescent cells induced by H2O2, radiation, or from aged mice would provide more information.

In the present study, the increased proliferation and osteogenesis of CD45-;CD31-;GFP- periskeletal progenitors were shown as paracrine mechanisms of *Chmp5*-deficient periskeletal progenitors to promote bone overgrowth in *Chmp5Ctsk* mice (Figs. 4F, G, and S4C-E). According to the reviewer’s suggestion, we have carried out the coculture experiment and the coculture of *Chmp5Ctsk* with *wild-type* skeletal progenitors could promote osteogenesis of *wild-type* cells (Fig. S4B), which further supports the paracrine effect of *Chmp5*-deficient periskeletal progenitors.

In addition, the cause and outcome of cell senescence could be highly heterogeneous, and different causes of cell senescence can cause significantly distinct, even opposite outcomes. Although the coculture experiments of WT periskeletal progenitors with senescent cells induced by H2O2, radiation, or from aged mice are very interesting, these are beyond the scope of the current study.

(3) Many EVs were secreted from Chmp5-deleted periskeletal progenitors, compared to the rarely detected EVs around WT cells. Since EVs of BMSCs or osteoprogenitors show strong effects of promoting osteogenesis, did the EVs contribute to the enhanced osteogenesis induced by Chmp5defeciency? Author’s response:

This is an interesting question. Although we did not separately test the effect of EVs from *Chmp5*-deficient periskeletal progenitors on the osteogenesis of WT skeletal progenitors, the CD45-;CD31-;GFP- skeletal progenitor cells from *Chmp5Ctsk* mice have an increased capacity of osteogenesis compared to corresponding cells from control animals (Figs. 4G and S4D). Also, the coculture of *Chmp5*-deficient with *wild-type* skeletal progenitors could enhance the osteogenesis of *wild-type* cells (Fig. S4B). These results suggest that EVs from *Chmp5*-deficient periskeletal progenitors could promote osteogenesis of neighboring WT skeletal progenitors. The specific functions of EVs of *Chmp5*-deficient periskeletal progenitors in regulating osteogenesis will be further investigated in future studies.

(4) EVs secreted from senescent cells propagate senescence and impair osteogenesis, why do EVs secreted from senescent cells induced by Chmp5-defeciency have opposite effects on osteogenesis?

The question is similar to comments #1 and #3 from this reviewer. First, the manifestations (including the secretory phenotype) and outcomes of cell senescence could be highly heterogeneous depending on inducers, tissue and cell contexts, and other factors such as “time”. Different causes of cell senescence could lead to different manifestations and outcomes, which have been discussed in the manuscript (Lines 381-383). Similarly, as mentioned above, skeletal stem/progenitor cells at different sites of musculoskeletal tissues could also demonstrate distinct, even opposite outcomes, as a result of cell senescence (Line 453-462). Second, CD45-;CD31-;GFP- periskeletal progenitor cells from *Chmp5Ctsk;Rosa2626mTmG/+* mice have an increased capacity of proliferation and osteogenesis compared to corresponding cells from control animals (Figs. 4F, G and S4C-E). Furthermore, the conditioned medium of *Chmp5*-deficient skeletal progenitors promoted the proliferation of ATDC5 cells (Fig. 4E) and the coculture of *Chmp5Ctsk* and *wild-type* periskeletal progenitors could enhance the osteogenesis of *wild-type* cells (Fig. S4B). Taken together, these results show paracrine actions of *Chmp5*-deficient periskeletal progenitors in promoting aberrant bone growth in *Chmp5* conditional knockout mice. We also refer the reviewer to our responses to comments #1 and #3.

(5) The Chmp5-ctsk mice show accelerated aging-related phenotypes, such as hair loss and joint stiffness. Did Ctsk also label cells in hair follicles or joint tissue?

This is an interesting question. Although we did not check the expression of CHMP5 in hair follicles, which is outside the scope of the present study, the result in Fig. 1E showed the expression of *Ctsk* in joint ligaments, tendons, and their insertion sites on the bone (Lines 108-111). Notably, the periskeletal bone overgrowth in *Chmp5Ctsk* mice was mainly mapped to insertion sites of ligaments and tendons on the bone, which have been discussed in the revised manuscript (Lines 456-460).

(6) Fifteen proteins were found to increase and five proteins to decrease in the cell supernatant of *Chmp5Ctsk* periskeletal progenitors. How about SASP factors in the secretory profile?

The SASP phenotype and related factors of senescent cells could be highly heterogeneous depending on inducers, cell types, and timing of senescence [10, 11]. Most of the proteins we identified in the secretome analysis have previously been reported in the secretory profile of osteoblasts or involved in the regulation of osteogenesis. Although we were interested in changes in common SASP factors, such as cytokines and chemokines, the experiment did not detect these factors, probably due to their small molecular weights and the technical limitations of the mass-spec analysis. We have clarified this in the revised manuscript. Specifically, we add the following sentences (Lines 258-261):

“Notably, the secretome analysis did not detect common SASP factors, such as cytokines and chemokines, in the secretory profile of *Chmp5Ctsk* periskeletal progenitors, probably due to their small molecular weights and the technical limitations of the mass-spec analysis.”

(7) D+Q treatment mitigates musculoskeletal pathologies in Chmp5 conditional knockout mice. In the previously published paper (CHMP5 controls bone turnover rates by dampening NF-κB activity in osteoclasts), inhibition of osteoclastic bone resorption rescues the aberrant bone phenotype of the Chmp5 conditional knockout mice. Whether the effects of D+Q on bone overgrowth is because of the inhibition of bone resorption?

This is an important question. We have discussed the potential off-target effect of senolytic drugs in *Chmp5Ctsk* mice in the revised manuscript. Specifically, we add the following paragraph (Lines 441451):

“Furthermore, it is unclear whether the effect of senolytic drugs in *Chmp5Ctsk* mice involves targeting osteoclasts other than osteogenic cells, as osteoclast senescence has not yet been evaluated. However, the efficacy of Q + D in targeting osteogenic cells, which is the focus of the current study, was confirmed in *Chmp5Dmp1* mice (Fig. 5C-E). Additionally, Q + D caused a higher cell apoptotic ratio in *Chmp5Ctsk* compared to *wild-type* periskeletal progenitors in ex vivo culture (Fig. 5A), demonstrating the effectiveness of Q + D in targeting osteogenic cells in the *Chmp5Ctsk* model. Furthermore, an alternative senolytic drug ABT-263 could also ameliorate periskeletal bone overgrowth in *Chmp5Ctsk* mice (Fig. 5F). Together, these results confirm that osteogenic cell senescence is responsible for the bone overgrowth in *Chmp5Ctsk* and *Chmp5Dmp1* mice and senolytic treatments are effective in alleviating these skeletal disorders.”

(8) The role of VPS4A in cell senescence should be measured to support the conclusion that CHMP5 regulates osteogenesis by affecting cell senescence.

We thank the reviewer for this suggestion. The current study mainly reports the function of CHMP5 in the regulation of skeletal progenitor cell senescence and osteogenesis. The roles of VPS4A in cell senescence and skeletal biology will be further explored in future studies. We have discussed this in the revised manuscript. Specifically, we add the following sentence (Lines 407-409):

“The roles of VPS4A in regulating musculoskeletal biology and cell senescence should be further explored in future studies.”

(9) Cell senescence with markers, such as p21 and H2AX, co-stained with GFP should be performed in the mouse models to indicate the effects of Chmp5 on cell senescence in vivo.

According to the reviewer’s suggestion, we have already performed immunostaining of γH2AX and colocalization with GFP in *Chmp5Ctsk;Rosa2626mTmG/+* and *Chmp5Ctsk/+;Rosa2626mTmG/+* mice. The results showed that there are more γH2AX+;GFP+ cells in the periskeletal overgrowth in *Chmp5Ctsk;Rosa2626mTmG/+* mice compared to the periosteum of *Chmp5Ctsk/+;Rosa2626mTmG/+* control animals. Because the γH2AX staining could stand as one of the critical results supporting the senescent phenotype of *Chmp5*-deficient periskeletal progenitors. We have added these results to Fig. 3E and put Fig. 3F in the original manuscript into Fig. S3E due to the space limitation in Figure 3. In sum, these results further enrich the senescent manifestations of *Chmp5*-deficient periskeletal progenitors.

(10) ADTC5 cell as osteochondromas cells line, is not a good cell model of periskeletal progenitors.Maybe primary periskeletal progenitor cell is a better choice.

ATDC5 cells are typically used as a chondrocyte progenitor cell line. However, our previous study showed that ATDC5 cells could also be used as a reasonable cell model for periskeletal progenitors [12], which was mentioned in the manuscript (Lines 202-204). In addition, the results of ATDC5 cells were also verified in primary periskeletal progenitor cells in this study.

**Recommendations for the authors:**

**Reviewer #1 (Recommendations for the authors):**
Despite the robust experimental framework and intriguing findings, there are several areas that require further attention to enhance the manuscript's overall quality and clarity:(1) The manuscript could benefit from a more in-depth discussion of the tissue-specific roles of CHMP5, particularly in addressing why CHMP5 deficiency results in distinct outcomes in osteogenic cells as opposed to other cell types, such as osteoclasts. Expanding the discussion would greatly enhance the comprehensiveness and clarity of the manuscript.

Based on the reviewer’s suggestion, we have expanded the discussion of the distinct roles of CHMP5 in different cell types. Specifically, we state (Lines 433-439):

“Also, a previous study by Adoro et al. did not detect endolysosomal abnormalities in _Chmp5_deficient developmental T cells [1]. Since both osteoclasts and T cells are of hematopoietic origin, and meanwhile osteogenic cells and MEFs, which show endolysosomal abnormalities after CHMP5 deficiency, are of mesenchymal origin, it turns out that the function of CHMP5 in regulating the endolysosomal pathway could be cell lineage-specific, which remains clarified in future studies.”

(2) Given that Figures 1 and 2 suggest that the absence of Chmp5 (CHMP5Ctsk & CHMP5Dmp1) leads to disordered proliferation or mineralization of bone or osteoblasts, the manuscript should delve deeper into the potential links between these findings and aging-related processes, such as age-associated fibrosis. Providing clearer explanations and discussion on these connections would help present a more cohesive understanding of the results in the context of aging.

We thank the reviewer for this favorable suggestion. A feature of aging is heterotopic ossification or mineralization in musculoskeletal soft tissues, including tendons and ligaments [7]. Notably, the abnormal periskeletal bone formation in *Chmp5Ctsk* mice in this study was mostly mapped to the insertion sites of tendons and ligaments on the bone (Fig. 1A and E), which is consistent with changes during aging and suggests that mechanical stress at these sites could be a contributor to periskeletal overgrowth. We have discussed these results in the revised manuscript. Specifically, we add the following paragraph (Lines 453-462):

“Notably, aging is associated with decreased osteogenic capacity in marrow stromal cells, which is related to conditions with low bone mass, such as osteoporosis. Rather, aging is also accompanied by increased ossification or mineralization in musculoskeletal soft tissues, such as tendons and ligaments [7]. In particular, the abnormal periskeletal overgrowth in *Chmp5Ctsk* mice was predominantly mapped to the insertion sites of tendons and ligaments on the bone (Fig. 1A and E), which is consistent with changes during aging and suggests that mechanical stress at these sites could contribute to the aberrant bone growth. These results suggest that skeletal stem/progenitor cells at different sites of musculoskeletal tissues could demonstrate different, even opposite outcomes in osteogenesis, due to cell senescence.”

(3) The manuscript would be improved by a more refined analysis in Figures 3 and 5, particularly in relation to the use of senolytic drugs. Furthermore, a detailed discussion of the specificity and potential off-target effects of quercetin and dasatinib treatments in Chmp5-deficient mice would strengthen the therapeutic claims of these drugs.

In Figure 3, we have added additional experiments and results to strengthen the senescent phenotypes of *Chmp5*-deficient periskeletal progenitors, including significant enrichment of the SAUL_SEN_MAYO geneset (positively correlated with cell senescence) and the KAMMINGA_SENESCENCE geneset (negatively correlated with cell senescence) at the transcriptional level by GSEA analysis of RNA-seq data (Fig. S3F), and an increase of γH2AX+;GFP+ cells at the site of periskeletal overgrowth in *Chmp5Ctsk;Rosa2626mTmG/+* mice compared to the periosteum of *Chmp5Ctsk/+;Rosa2626mTmG/+* control mice (Fig. 3E). These results further enrich the senescent molecular manifestations of *Chmp5*-deficient periskeletal progenitors.

In Figure 5, we used an alternative senolytic drug ABT-263 to treat *Chmp5Ctsk* mice, and this antisenescence treatment could also alleviate periskeletal bone overgrowth in this mouse model (Fig. 5F). Furthermore, we have also discussed the potential off-target effects of senolytic drugs in *Chmp5Ctsk* mice. Specifically, we add the following paragraph (Lines 441-451):

“Furthermore, it is unclear whether the effect of senolytic drugs in *Chmp5Ctsk* mice involves targeting osteoclasts other than osteogenic cells, as osteoclast senescence has not yet been evaluated. However, the efficacy of Q + D in targeting osteogenic cells, which is the focus of the current study, was confirmed in *Chmp5Dmp1* mice (Fig. 5C-E). Additionally, Q + D caused a higher cell apoptotic ratio in *Chmp5Ctsk* compared to *wild-type* periskeletal progenitors in ex vivo culture (Fig. 5A), demonstrating the effectiveness of Q + D in targeting osteogenic cells in the *Chmp5Ctsk* model. Furthermore, an alternative senolytic drug ABT-263 could also ameliorate periskeletal bone overgrowth in *Chmp5Ctsk* mice (Fig. 5F). Together, these results confirm that osteogenic cell senescence is responsible for the bone overgrowth in *Chmp5Ctsk* and *Chmp5Dmp1* mice and senolytic treatments are effective in alleviating these skeletal disorders.”

(4) The manuscript could be further enhanced by providing more details into how CHMP5 specifically regulates VPS4A protein levels. Notably, this is a central aspect of the paper linking CHMP5 to endolysosomal dysfunction.

We thank the reviewer for this important suggestion. One of the novel findings of this study is that CHMP5 regulates the protein level of VPS4A without affecting its RNA transcription. The mechanism of CHMP5 in the regulation of VPS4A protein will be reported in a separate study. However, we have discussed the potential mechanism in the manuscript (Lines 399-409). Specifically, we state:

“However, the mechanism of CHMP5 in the regulation of the VPS4A protein has not yet been studied. Since CHMP5 can recruit the deubiquitinating enzyme USP15 to stabilize IκBα in osteoclasts by suppressing ubiquitination-mediated proteasomal degradation [13], it is also possible that CHMP5 stabilizes the VPS4A protein by recruiting deubiquitinating enzymes and regulating the ubiquitination of VPS4A, which needs to be clarified in future studies. Notably, mutations in the *VPS4A* gene in humans can cause multisystemic diseases, including musculoskeletal abnormalities [14] (OMIM: 619273), suggesting that normal expression and function of VPS4A are important for musculoskeletal physiology. The roles of VPS4A in regulating musculoskeletal biology and cell senescence should be further explored in future studies.”

(5) The discussion section could be enriched by more thoroughly integrating the current findings with previous studies on CHMP5, particularly those exploring its role in osteoclast differentiation and NF-κB signaling.

The comment is similar to comment #1 of this reviewer. We have expanded the discussion of the distinct functions of CHMP5 in osteoclasts and osteogenic cells (Lines 424-439). We also refer the reviewer to our response to comment #1.

(6) Figure S4 D is incorrectly arranged and should be revised accordingly.

Sorry for the confusion. We have added additional annotations to make the images clearer. Now it is Fig. S4E in the revised manuscript.

**Reviewer #2 (Recommendations for the authors):**
(1) Abstract A clinical perspective or at least an outline is desirable.

The clinical importance of the findings of this study in understanding and treating musculoskeletal disorders of lysosomal storage diseases has been highlighted at the end of the abstract (Line 38).

(2) Introduction Header missing.The protein name is BCL2, not Bcl2.

These have been corrected in the revised manuscript (Lines 41, 66).

(3) ResultsThe mouse phenotype experiments are well done.Hmga1, Hmga2, Trp53, Ets1, and Txn1 are no typical senescence-associated genes. How aboutCdkn2a and Cdkn1a? These could easily be highlighted in Figure 3B.

Hmga1, Hmga2, Trp53, Ets1, and Txn1 are within the geneset of Reactome Cellular Senescence. Notably, only the protein levels of CDKN2A (p16) and CDKN1A (p21) showed significant changes (Fig. 3D) and the mRNA levels of *Cdkn2a* and *Cdkn1a* did not show significant changes according to RNAseq data. We have added the result of *Cdkn2a* and *Cdkn1a* mRNA levels to Fig. S3D in the revised manuscript. Also, we add the following sentences in the text (Lines 193-195):

“However, the mRNA levels of *Cdkn2a* (p16) and *Cdkn1a* (p21) did not show significant changes according to the RNA-seq analysis (Fig. S3D).”

Figure 3C: Which gene set was used for SASP?

The SASP geneset in Fig. 3C was from the Reactome database. We have clarified this in the figure legend of Fig. 3 in the revised manuscript (Line 1013).

The symptom "joint stiffness/contracture" could also be due to skeletal abnormalities related to Chmp5Ctsk.

Joint stiffness/contracture during aging is mainly the result of heterotopic ossification or mineralization in musculoskeletal soft tissues, including ligaments, tendons, joint capsules, and their insertion sites on the bone. Notably, the periskeletal bone overgrowth in *Chmp5Ctsk* mice was mainly mapped to the insertion sites of tendons, ligaments, and joint capsules on the bone, which are consistent with changes during aging. These results have been discussed in the revised manuscript (Lines 456-460).

Overall, cellular senescence needs at least Cdkn2a and/or Cdkn1a and another marker, i.e. SenMayo or telomere-associated foci or senescence-associated distortion of satellites.

We have run GSEA with the SenMayo geneset and the result is added in Fig. S3F in the revised manuscript. Also, we ran another geneset KAMMINGA_SENESCENCE which includes genes downregulated in cell senescence. Both genesets are significantly enriched in *Chmp5*-deficient periskeletal progenitors based on RNA-seq data (Fig. S3F).

In addition, we also performed immunostaining for another senescence marker γH2AX and the results showed that there are more γH2AX+;GFP+ cells in periskeletal overgrowth in *Chmp5Ctsk;Rosa26mTmG/+* mice compared to the periosteum of *Chmp5Ctsk/+;Rosa2626mTmG/+* control animals (Fig. 3E).

Together, these results further support the senescent phenotypes of *Chmp5*-deficient periskeletal progenitors.

For Figure 4A: What is the NES?

The value of NES has been added in Fig. 4A.

The existence of vesicles does not necessarily indicate more SASP. Author’s response:

We agree with the reviewer that the secretion of extracellular vesicles is not directly correlated with the SASP. In this study, the increased secretory vesicles around *Chmp5Ctsk* periskeletal progenitors represent a secretory phenotype of *Chmp5*-deficient periskeletal progenitors and have paracrine effects in the abnormal bone growth in *Chmp5* conditional knockout mice as shown in Figs. 4 and S4.

The Chmp5-deficient cells COULD promote the proliferation and osteogenesis of other progenitors, but they might as well not. And if this is through the SASP, is completely unresolved.

CD45^-^;CD31^-^;GFP^-^ periskeletal progenitor cells from *Chmp5Ctsk;Rosa2626mTmG/+* mice showed an increased capacity of proliferation and osteogenesis compared to the corresponding cells from control animals (Figs. 4F, G, and S4C-E). Also, the conditioned medium of *Chmp5*-deficient skeletal progenitors promoted the proliferation of ATDC5 cells (Fig. 4E). In addition, the coculture of *Chmp5Ctsk* and *wild-type* periskeletal progenitors could enhance the osteogenesis of *wild-type* cells (Fig. S4B). These results demonstrate the paracrine actions of *Chmp5*-deficient periskeletal progenitors in promoting aberrant bone growth in *Chmp5Ctsk* and *Chmp5Dmp1* mice. However, factors that mediate the paracrine effects of *Chmp5*-deficient periskeletal progenitors remain further clarified in future studies.

This has been mentioned in the revised manuscript (Lines 263-265).

Figure 5C: The time points are not labelled.

The time point of 16 weeks was mentioned in the Method section and now it has been added in the legend of Fig. 5C (Line 1063).

Figure B: Was the bone's overall thickness quantified?

In Fig. 5B, bone morphology in *Chmp5Ctsk* mice is irregular and difficult to quantify. Therefore, we did not qualify the overall bone thickness in these animals. However, the thickness of the cortical bone was measured by micro-CT analysis in *Chmp5Dmp1* mice after treatment with Q + D (Fig. 5E). Also, we have added the image of the gross femur thickness of *Chmp5Dmp1* mice before and after treatment with Q + D in Fig. 5E.

It needs to be demonstrated that the actual cell number was reduced after D+Q treatment.

The Q + D treatment caused a higher cell apoptotic ratio in *Chmp5Ctsk* vs. *wild-type* skeletal progenitors in ex vivo culture (Fig. 5A), suggesting its effectiveness in targeting the senescent periskeletal progenitors.

Figure 7A: What is the NES?

The value of NES has been added in Fig. 7A.

**Reviewer #3 (Recommendations for the authors):**
(1) The WB analysis should be quantified in the Figure 3D.

In Fig. 3D, the numbers above the lanes of p16 and p21 are the results of the quantification of the band intensity after normalization by β-Actin, which has been indicated in the Figure legend (Lines 10151017).

(2) The osteoblast detection should be measured with antibody against osteocalcin.

This comment did not specify what result the reviewer was referring to. However, most of the experiments in this study were performed in primary skeletal progenitor cells or cell lines. Osteoblasts were not specifically involved in the current study.

(3) Co-culture of Chmp5-KO periskeletal progenitors with WT ones should be conducted to detect the migration and osteogenesis of WT cell in response to Chmp5-KO induced senescent cells. In addition, co-culture of WT periskeletal progenitors with senescent cells induced by H2O2, radiation, or from aged mice would provide more information.

This comment is the same as comment #2 in the Public Reviews of this Reviewer. We already carried out the coculture experiment of *Chmp5*-deficient and *wild-type* periskeletal progenitors and the result was added in Fig. S4B. We refer the reviewer to our response to comment #2 in the Public Reviews for more details.

(4) D+Q treatment mitigates musculoskeletal pathologies in Chmp5 conditional knockout mice. In the previously published paper (CHMP5 controls bone turnover rates by dampening NF-κB activity in osteoclasts), inhibition of osteoclastic bone resorption rescues the aberrant bone phenotype of the Chmp5 conditional knockout mice. Is the effect of D+Q on bone overgrowth because of the inhibition of bone resorption?

This comment is the same as comment #7 in the Public Reviews of this Reviewer, where we already address this question.

(5) The role of VPS4A in cell senescence should be measured to support the conclusion that CHMP5 regulates osteogenesis through affecting cell senescence.

This comment is the same as comment #8 in the Public Reviews of this Reviewer. We refer the reviewer to our response to that comment.

(6) Cell senescence with the markers, such as p21 and H2AX, co-stained with GFP should be performed in the mouse models to indicate the effects of Chmp5 on cell senescence in vivo.

This comment is the same as comment #9 in the Public Reviews of this Reviewer. We have performed immunostaining of γH2AX and colocalization with GFP in *Chmp5Ctsk;Rosa2626mTmG/+* mice and *Chmp5Ctsk/+;Rosa2626mTmG/+* mice. The results showed that there were more γH2AX+;GFP+ cells at the site of periskeletal overgrowth in *Chmp5Ctsk;Rosa2626mTmG/+* mice compared to the periosteum of *Chmp5Ctsk/+;Rosa2626mTmG/+* control mice (Fig. 3E). We also refer the reviewer to our response to comment #9 in Public Reviews.

(7) ADTC5 cell as osteochondromas cells line, is not a good cell model of periskeletal progenitors.Maybe primary periskeletal progenitor cell is a better choice.

This comment is the same as comment #10 in the Public Reviews of this Reviewer. Our previous study showed that ATDC5 cells could be used as a reasonable cell model for periskeletal progenitors [12]. Also, most of the results of ATDC5 cells in the current study were verified in primary periskeletal progenitors.

References

(1) Adoro S, Park KH, Bettigole SE, Lis R, Shin HR, Seo H, et al. Post-translational control of T cell development by the ESCRT protein CHMP5. Nat Immunol. 2017;18(7):780-90. doi: 10.1038/ni.3764. PubMed PMID: 28553951.

(2) Kassem M, Bianco P. Skeletal stem cells in space and time. Cell. 2015;160(1-2):17-9. doi: 10.1016/j.cell.2014.12.034. PubMed PMID: 25594172.

(3) Chan CKF, Gulati GS, Sinha R, Tompkins JV, Lopez M, Carter AC, et al. Identification of the Human Skeletal Stem Cell. Cell. 2018;175(1):43-56 e21. doi: 10.1016/j.cell.2018.07.029. PubMed PMID: 30241615.

(4) Debnath S, Yallowitz AR, McCormick J, Lalani S, Zhang T, Xu R, et al. Discovery of a periosteal stem cell mediating intramembranous bone formation. Nature. 2018;562(7725):133-9. Epub 20180924. doi: 10.1038/s41586-018-0554-8. PubMed PMID: 30250253; PubMed Central PMCID: PMCPMC6193396.

(5) Mizuhashi K, Ono W, Matsushita Y, Sakagami N, Takahashi A, Saunders TL, et al. Resting zone of the growth plate houses a unique class of skeletal stem cells. Nature. 2018;563(7730):254-8. doi: 10.1038/s41586-018-0662-5. PubMed PMID: 30401834; PubMed Central PMCID: PMCPMC6251707.

(6) Zhang F, Wang Y, Zhao Y, Wang M, Zhou B, Zhou B, et al. NFATc1 marks articular cartilage progenitors and negatively determines articular chondrocyte differentiation. Elife. 2023;12. Epub 20230215. doi: 10.7554/eLife.81569. PubMed PMID: 36790146; PubMed Central PMCID: PMCPMC10076019.

(7) Dai GC, Wang H, Ming Z, Lu PP, Li YJ, Gao YC, et al. Heterotopic mineralization (ossification or calcification) in aged musculoskeletal soft tissues: A new candidate marker for aging. Ageing Res Rev. 2024;95:102215. Epub 20240205. doi: 10.1016/j.arr.2024.102215. PubMed PMID: 38325754.

(8) Mohler ER, 3rd, Adam LP, McClelland P, Graham L, Hathaway DR. Detection of osteopontin in calcified human aortic valves. Arterioscler Thromb Vasc Biol. 1997;17(3):547-52. doi: 10.1161/01.atv.17.3.547. PubMed PMID: 9102175.

(9) Mohler ER, 3rd, Gannon F, Reynolds C, Zimmerman R, Keane MG, Kaplan FS. Bone formation and inflammation in cardiac valves. Circulation. 2001;103(11):1522-8. doi: 10.1161/01.cir.103.11.1522. PubMed PMID: 11257079.

(10) Paramos-de-Carvalho D, Jacinto A, Saude L. The right time for senescence. Elife. 2021;10. Epub 2021/11/11. doi: 10.7554/eLife.72449. PubMed PMID: 34756162; PubMed Central PMCID: PMCPMC8580479.

(11) Wiley CD, Campisi J. The metabolic roots of senescence: mechanisms and opportunities for intervention. Nat Metab. 2021;3(10):1290-301. Epub 2021/10/20. doi: 10.1038/s42255-021-00483-8. PubMed PMID: 34663974; PubMed Central PMCID: PMCPMC8889622.

(12) Ge X, Tsang K, He L, Garcia RA, Ermann J, Mizoguchi F, et al. NFAT restricts osteochondroma formation from entheseal progenitors. JCI Insight. 2016;1(4):e86254. doi: 10.1172/jci.insight.86254. PubMed PMID: 27158674; PubMed Central PMCID: PMCPMC4855520.

(13) Greenblatt MB, Park KH, Oh H, Kim JM, Shin DY, Lee JM, et al. CHMP5 controls bone turnover rates by dampening NF-kappaB activity in osteoclasts. J Exp Med. 2015;212(8):1283-301. Epub 20150720. doi: 10.1084/jem.20150407. PubMed PMID: 26195726; PubMed Central PMCID: PMCPMC4516796.

(14) Rodger C, Flex E, Allison RJ, Sanchis-Juan A, Hasenahuer MA, Cecchetti S, et al. De Novo VPS4A Mutations Cause Multisystem Disease with Abnormal Neurodevelopment. Am J Hum Genet. 2020;107(6):1129-48. Epub 20201112. doi: 10.1016/j.ajhg.2020.10.012. PubMed PMID: 33186545; PubMed Central PMCID: PMCPMC7820634.